# Generalizing Importance Weighting to A Universal Solver for Distribution Shift Problems

**Tongtong Fang**[1]    **Nan Lu**[2,3]    **Gang Niu**[3*]    **Masashi Sugiyama**[3,1]
[1]The University of Tokyo, Japan    [2]University of Tübingen, Germany    [3]RIKEN, Japan

## Abstract

*Distribution shift* (DS) may have two levels: the distribution itself changes, and the *support* (i.e., the set where the probability density is non-zero) also changes. When considering the support change between the training and test distributions, there can be four cases: (i) they exactly match; (ii) the training support is wider (and thus covers the test support); (iii) the test support is wider; (iv) they partially overlap. Existing methods are good at cases (i) and (ii), while cases (iii) and (iv) are more common nowadays but still under-explored. In this paper, we generalize *importance weighting* (IW), a golden solver for cases (i) and (ii), to a universal solver for all cases. Specifically, we first investigate why IW might fail in cases (iii) and (iv); based on the findings, we propose *generalized IW* (GIW) that could handle cases (iii) and (iv) and would reduce to IW in cases (i) and (ii). In GIW, the test support is split into an *in-training* (IT) part and an *out-of-training* (OOT) part, and the *expected risk* is decomposed into a weighted classification term over the IT part and a standard classification term over the OOT part, which guarantees the *risk consistency* of GIW. Then, the implementation of GIW consists of three components: (a) the split of validation data is carried out by the one-class support vector machine, (b) the first term of the *empirical risk* can be handled by any IW algorithm given training data and IT validation data, and (c) the second term just involves OOT validation data. Experiments demonstrate that GIW is a universal solver for DS problems, outperforming IW methods in cases (iii) and (iv).

## 1 Introduction

Deep *supervised classification* has been successful where the training and test data should come from the same distribution (Goodfellow et al., 2016). When this assumption does not hold in practice, we suffer from *distribution shift* (DS) problems and the learned classifier may often generalize poorly (Quionero-Candela et al., 2009; Pan and Yang, 2009; Sugiyama and Kawanabe, 2012). Let $\boldsymbol{x}$ and $y$ be the instance (i.e., input) and class-label (i.e., output) random variables. Then, DS means that the *underlying joint density* of the training data $p_{\mathrm{tr}}(\boldsymbol{x}, y)$ differs from that of the test data $p_{\mathrm{te}}(\boldsymbol{x}, y)$.

There are two levels in the DS research. At the first level, only the change of the data distribution is considered. With additional assumptions, DS can be reduced into covariate shift $p_{\mathrm{tr}}(\boldsymbol{x}) \neq p_{\mathrm{te}}(\boldsymbol{x})$, class-prior shift $p_{\mathrm{tr}}(y) \neq p_{\mathrm{te}}(y)$, class-posterior shift $p_{\mathrm{tr}}(y \mid \boldsymbol{x}) \neq p_{\mathrm{te}}(y \mid \boldsymbol{x})$, and class-conditional shift $p_{\mathrm{tr}}(\boldsymbol{x} \mid y) \neq p_{\mathrm{te}}(\boldsymbol{x} \mid y)$ (Quionero-Candela et al., 2009). We focus on joint shift $p_{\mathrm{tr}}(\boldsymbol{x}, y) \neq p_{\mathrm{te}}(\boldsymbol{x}, y)$, as it is the most general and difficult case of DS. At the second level, the change of the *support* of the data distribution is also considered, where given any joint density $p(\boldsymbol{x}, y)$, its support is defined as the set $\{(\boldsymbol{x}, y) : p(\boldsymbol{x}, y) > 0\}$. More specifically, denote by $\mathcal{S}_{\mathrm{tr}}$ and $\mathcal{S}_{\mathrm{te}}$ the support of $p_{\mathrm{tr}}(\boldsymbol{x}, y)$ and $p_{\mathrm{te}}(\boldsymbol{x}, y)$, respectively. When considering the relationship between $\mathcal{S}_{\mathrm{tr}}$ and $\mathcal{S}_{\mathrm{te}}$, there can be four cases:

(i) $\mathcal{S}_{\mathrm{tr}}$ and $\mathcal{S}_{\mathrm{te}}$ exactly match, i.e., $\mathcal{S}_{\mathrm{tr}} = \mathcal{S}_{\mathrm{te}}$;

---

*Correspondence to: GN <gang.niu.ml@gmail.com>, TF <fang@ms.k.u-tokyo.ac.jp>

37th Conference on Neural Information Processing Systems (NeurIPS 2023).

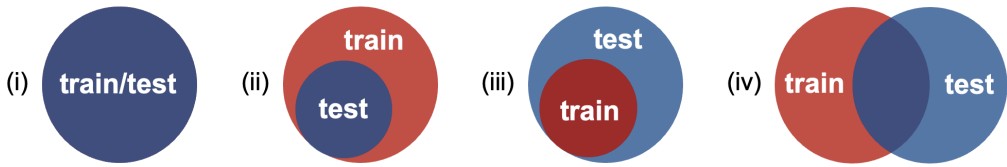

Figure 1: An illustration of the relationship between the training support and the test support.

(ii) $\mathcal{S}_{\mathrm{tr}}$ is wider and covers $\mathcal{S}_{\mathrm{te}}$, i.e., $\mathcal{S}_{\mathrm{tr}} \supset \mathcal{S}_{\mathrm{te}}$ and $\mathcal{S}_{\mathrm{tr}} \setminus \mathcal{S}_{\mathrm{te}} \neq \emptyset$;

(iii) $\mathcal{S}_{\mathrm{te}}$ is wider and covers $\mathcal{S}_{\mathrm{tr}}$, i.e., $\mathcal{S}_{\mathrm{tr}} \subset \mathcal{S}_{\mathrm{te}}$ and $\mathcal{S}_{\mathrm{te}} \setminus \mathcal{S}_{\mathrm{tr}} \neq \emptyset$;

(iv) $\mathcal{S}_{\mathrm{tr}}$ and $\mathcal{S}_{\mathrm{te}}$ partially overlap, i.e., $\mathcal{S}_{\mathrm{tr}} \cap \mathcal{S}_{\mathrm{te}} \neq \emptyset$, $\mathcal{S}_{\mathrm{tr}} \setminus \mathcal{S}_{\mathrm{te}} \neq \emptyset$, and $\mathcal{S}_{\mathrm{te}} \setminus \mathcal{S}_{\mathrm{tr}} \neq \emptyset$.[1]

The four cases are illustrated in Figure 1. We focus on cases (iii) and (iv), as they are more general and more difficult than cases (i) and (ii).

**Problem setting** Denote by $\mathcal{X}$ and $\mathcal{Y}$ the input and output domains, where $\mathcal{Y} = \{1, \ldots, C\}$ for $C$-class classification problems. Let $\boldsymbol{f} : \mathcal{X} \to \mathbb{R}^C$ be a classifier (to be trained) and $\ell : \mathbb{R}^C \times \mathcal{Y} \to (0, +\infty)$ be a loss function (for training $\boldsymbol{f}$).[2] Then, the *risk* is defined as follows (Vapnik, 1998):

$$R(\boldsymbol{f}) = \mathbb{E}_{p_{\mathrm{te}}(\boldsymbol{x}, y)}[\ell(\boldsymbol{f}(\boldsymbol{x}), y)], \tag{1}$$

where $\mathbb{E}[\cdot]$ denotes the expectation. In the joint-shift problems, we are given a training set $\mathcal{D}_{\mathrm{tr}} = \{(\boldsymbol{x}_i^{\mathrm{tr}}, y_i^{\mathrm{tr}})\}_{i=1}^{n_{\mathrm{tr}}} \overset{\mathrm{i.i.d.}}{\sim} p_{\mathrm{tr}}(\boldsymbol{x}, y)$ and a validation set $\mathcal{D}_{\mathrm{v}} = \{(\boldsymbol{x}_i^{\mathrm{v}}, y_i^{\mathrm{v}})\}_{i=1}^{n_{\mathrm{v}}} \overset{\mathrm{i.i.d.}}{\sim} p_{\mathrm{te}}(\boldsymbol{x}, y)$, where $\mathcal{D}_{\mathrm{tr}}$ is much bigger than $\mathcal{D}_{\mathrm{v}}$, i.e., $n_{\mathrm{tr}} \gg n_{\mathrm{v}}$. The goal is to reliably estimate the risk from $\mathcal{D}_{\mathrm{tr}}$ and $\mathcal{D}_{\mathrm{v}}$ and train $\boldsymbol{f}$ by minimizing the empirical risk, which should outperform training $\boldsymbol{f}$ from only $\mathcal{D}_{\mathrm{v}}$.

**Motivation** *Importance weighting* (IW) has been a golden solver for DS problems (Sugiyama and Kawanabe, 2012), and there are many great off-the-shelf IW methods (Huang et al., 2007; Sugiyama et al., 2007a,b; Kanamori et al., 2009). Recently, *dynamic IW* (DIW) was proposed to make IW compatible with stochastic optimizers and thus it can be used for deep learning (Fang et al., 2020). However, all IW methods including DIW have assumed cases (i) and (ii)—in cases (iii) and (iv), IW methods become problematic. Specifically, as the importance weights are only used on $\mathcal{S}_{\mathrm{tr}}$, even though they become ill-defined on $\mathcal{S}_{\mathrm{te}} \setminus \mathcal{S}_{\mathrm{tr}}$, IW itself is still well-defined. Nevertheless, in such a situation, the IW *identity* will become an *inequality* (i.e., Theorem 2), which means that what we minimize for training is no longer an approximation of the original risk $R(\boldsymbol{f})$ and thus IW may lead to poor trained classifiers (i.e., Proposition 3). Moreover, some IW-like methods based on bilevel optimization share a similar issue with IW (Jiang et al., 2018; Ren et al., 2018; Shu et al., 2019), since $\boldsymbol{f}$ is only trained from $\mathcal{D}_{\mathrm{tr}}$ where $\mathcal{D}_{\mathrm{v}}$ is used to determine the importance weights on $\mathcal{D}_{\mathrm{tr}}$. In fact, cases (iii) and (iv) are more common nowadays due to *data-collection biases*, but they are still under-explored. For example, a class has several subclasses, but not all subclasses are presented in $\mathcal{D}_{\mathrm{tr}}$ (see Figure 2). Therefore, we want to generalize IW to a universal solver for all the four cases.

**Contributions** Our contributions can be summarized as follows.

- Firstly, we theoretically and empirically analyze when and why IW methods can succeed/may fail. We reveal that the objective of IW is good in cases (i) and (ii) and bad in cases (iii) and (iv).
- Secondly, we propose *generalized IW* (GIW). In GIW, $\mathcal{S}_{\mathrm{te}}$ is split into an *in-training* (IT) part $\mathcal{S}_{\mathrm{te}} \cap \mathcal{S}_{\mathrm{tr}}$ and an *out-of-training* (OOT) part $\mathcal{S}_{\mathrm{te}} \setminus \mathcal{S}_{\mathrm{tr}}$, and its objective consists of a weighted classification term over the IT part and a standard classification term over the OOT part. GIW is justified as its objective is good in all the four cases and reduces to IW in cases (i) and (ii). Thus, GIW is a *strict generalization* of IW from the objective point of view, and GIW is safer to be used when we are not sure whether the problem to be solved is a good case or a bad case for IW.[3]
- Thirdly, we provide a practical implementation of GIW: (a) following the split of $\mathcal{S}_{\mathrm{te}}$, $\mathcal{D}_{\mathrm{v}}$ is split into an IT set and an OOT set using the *one-class support vector machine* (Schölkopf et al., 1999);

---

[1]When $\mathcal{S}_{\mathrm{tr}}$ and $\mathcal{S}_{\mathrm{te}}$ differ, they differ by a non-zero probability measure; otherwise, we regard it as case (i). For example, $\mathcal{S}_{\mathrm{te}} \setminus \mathcal{S}_{\mathrm{tr}} \neq \emptyset$ in cases (iii) and (iv) means that $\sum_y \int_{\{\boldsymbol{x} : (\boldsymbol{x}, y) \in \mathcal{S}_{\mathrm{te}} \setminus \mathcal{S}_{\mathrm{tr}}\}} p_{\mathrm{te}}(\boldsymbol{x}, y) \mathrm{d}\boldsymbol{x} > 0$.

[2]The positivity of $\ell$, i.e., $\ell(\boldsymbol{f}(\boldsymbol{x}), y) > 0$ rather than $\ell(\boldsymbol{f}(\boldsymbol{x}), y) \geq 0$, is needed to prove Theorem 2. This assumption holds for the popular *cross-entropy loss* and its robust variants, since $\boldsymbol{f}(\boldsymbol{x})$ must stay finite.

[3]Even though we have divided all DS problems into four cases according to support shift (SS), we cannot empirically detect tiny or huge SS (under joint shift). Given that there is already DS, existing *two-sample test* methods for detecting DS are not good at detecting whether there is SS or not—their results must be positive.

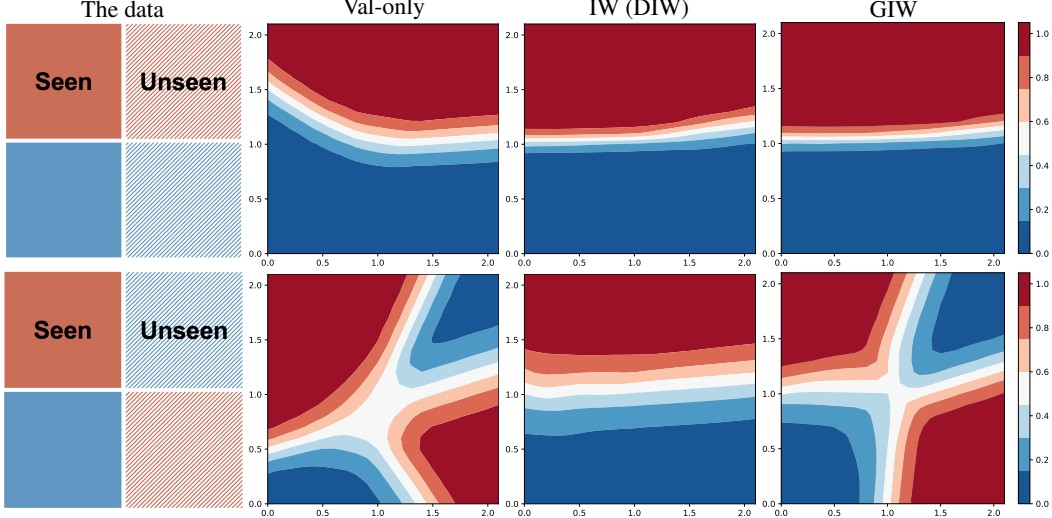

This is binary classification to distinguish red and blue synthetic data that are uniformly distributed in a 2-by-2 grid (consisting of 4 squares in different positions). The training distribution $p_{\mathrm{tr}}(\boldsymbol{x}, y)$ includes only the left 2 squares, and the test distribution $p_{\mathrm{te}}(\boldsymbol{x}, y)$ includes all the 4 squares; there are only 4 validation data, 1 in each square. "Val-only" means using only the validation data to train the model, "DIW" refers to Fang et al. (2020), and "GIW" is the proposed method. The learned decision boundaries are plotted to compare those methods. In the top panel, IW and GIW perform well, but in the bottom panel, IW completely fails and GIW still performs well. The performance of Val-only is not satisfactory since there are too few validation data. More details and discussions about this experiment can be found in the last part of Section 2.

Figure 2: Two concrete examples of the success and failure of IW in case (iii).

(b) the IT set, instead of the whole $\mathcal{D}_{\mathrm{v}}$, is used for IW; and (c) the OOT set directly joins training together with $\mathcal{D}_{\mathrm{tr}}$ since no data in $\mathcal{D}_{\mathrm{tr}}$ comes from the OOT part.

• Finally, we design and conduct extensive experiments that demonstrate the effectiveness of GIW in cases (iii) and (iv). The experiment design is also a major contribution since no experimental setup is available for reference to simulate case (iii) or (iv) on benchmark datasets.

**Organization** The analyses of IW are in Section 2, the proposal of GIW is in Section 3, and the experiments are in Section 4. Related work and additional experiments are in the appendices.

## 2 A deeper understanding of IW

First, we review the traditional *importance weighting* (IW) and its modern implementation *dynamic importance weighting* (DIW). Then, we analyze when and why IW methods can succeed/may fail.

**A review of IW** Let $w^*(\boldsymbol{x}, y) = p_{\mathrm{te}}(\boldsymbol{x}, y)/p_{\mathrm{tr}}(\boldsymbol{x}, y)$, which is the ratio of the test density $p_{\mathrm{te}}(\boldsymbol{x}, y)$ over the training density $p_{\mathrm{tr}}(\boldsymbol{x}, y)$, known as the *importance function*. Then, the *expected objective* of IW can be expressed as

$$J(\boldsymbol{f}) = \mathbb{E}_{p_{\mathrm{tr}}(\boldsymbol{x}, y)}[w^*(\boldsymbol{x}, y)\ell(\boldsymbol{f}(\boldsymbol{x}), y)]. \tag{2}$$

In order to empirically approximate $J(\boldsymbol{f})$ in (2), we need to have an empirical version $\widehat{w}(\boldsymbol{x}, y)$ of $w^*(\boldsymbol{x}, y)$, so that the *empirical objective* of IW is[4]

$$\widehat{J}(\boldsymbol{f}) = \frac{1}{n_{\mathrm{tr}}} \sum_{i=1}^{n_{\mathrm{tr}}} \widehat{w}(\boldsymbol{x}_i^{\mathrm{tr}}, y_i^{\mathrm{tr}})\ell(\boldsymbol{f}(\boldsymbol{x}_i^{\mathrm{tr}}), y_i^{\mathrm{tr}}). \tag{3}$$

The original IW method is implemented in two steps: (I) *weight estimation* (WE) where $\widehat{w}(\boldsymbol{x}, y)$ is obtained and (II) *weighted classification* (WC) where $\widehat{J}(\boldsymbol{f})$ is minimized. The first step relies on the training data $\mathcal{D}_{\mathrm{tr}}$ and the validation data $\mathcal{D}_{\mathrm{v}}$, and it can be either estimating the two density functions separately and taking their ratio or directly estimating the density ratio (Sugiyama et al., 2012).

---

[4]As an empirical risk estimator, $\widehat{J}(\boldsymbol{f})$ in Eq. (3) is *not unbiased* to $J(\boldsymbol{f})$ in Eq. (2), since $\widehat{w}(\boldsymbol{x}, y)$ is not unbiased to $w^*(\boldsymbol{x}, y)$. As far as we know, in IW, there is no unbiased importance-weight estimator and thus no unbiased risk estimator. That being said, $\widehat{J}(\boldsymbol{f})$ can still be *(statistically) consistent* with $J(\boldsymbol{f})$ under mild conditions if $\widehat{w}(\boldsymbol{x}, y)$ is consistent with $w^*(\boldsymbol{x}, y)$.

**A review of DIW** The aforementioned two-step approach is very nice when the classifier $\boldsymbol{f}$ is a simple model, but it has a serious issue when $\boldsymbol{f}$ is a deep model (Fang et al., 2020). Since WE is not equipped with representation learning, in order to boost its expressive power, we need an external feature extractor such as an internal representation learned by WC. As a result, we are trapped by a *circular dependency*: originally we need $w^*$ to train $\boldsymbol{f}$; now we need a trained $\boldsymbol{f}$ to estimate $w^*$.

DIW (Fang et al., 2020) has been proposed to resolve the critical circular dependency and to make IW usable for deep learning. Specifically, DIW uses a non-linear transformation $\pi$ created from the current $\boldsymbol{f}$ (being trained) and replaces $w^*(\boldsymbol{x}, y)$ with $w^*(\boldsymbol{z}) = p_{\text{te}}(\boldsymbol{z})/p_{\text{tr}}(\boldsymbol{z})$, where $\boldsymbol{z} = \pi(\boldsymbol{x}, y)$ is the current *loss-value* or *hidden-layer-output* representation of $(\boldsymbol{x}, y)$. DIW iterates between WE for estimating $w^*(\boldsymbol{z})$ and WC for training $\boldsymbol{f}$ and thus updating $\pi$ in a seamless mini-batch-wise manner. Given that WE enjoys representation learning inside WC, the importance-weight estimation quality of WE and the classifier training quality of WC can improve each other gradually but significantly.

**Risk consistency/inconsistency of IW** Now, consider how to qualify good or bad expected objectives under different conditions. To this end, we adopt the concepts of *risk consistency* and *classifier consistency* from the label-noise learning literature (Xia et al., 2019, 2020; Yao et al., 2020).

**Definition 1.** Given an (expected) objective $J(\boldsymbol{f})$, we say it is *risk-consistent* if $J(\boldsymbol{f}) = R(\boldsymbol{f})$ for any $\boldsymbol{f}$, i.e., the objective is equal to the original risk for any classifier. On the other hand, we say $J(\boldsymbol{f})$ is *classifier-consistent* if $\arg\min_{\boldsymbol{f}} J(\boldsymbol{f}) = \arg\min_{\boldsymbol{f}} R(\boldsymbol{f})$ where the minimization is taken over all measurable functions, i.e., the objective shares the optimal classifier with the original risk.

In the definition above, risk consistency is conceptually stronger than classifier consistency. If an objective is risk-consistent, it must also be classifier-consistent; if it is classifier-consistent, it may sometimes be risk-inconsistent. Note that a risk-inconsistent objective is not necessarily very bad, as it can still be classifier-consistent.[5] Hence, when considering expected objectives, risk consistency is a sufficient condition and classifier consistency is a necessary condition for good objectives.

In what follows, we analyze when and why the objective of IW, namely $J(\boldsymbol{f})$ in (2), can be a good objective or may be a bad objective.

**Theorem 1.** *In cases (i) and (ii), IW is risk-consistent.*[6]
*Proof.* Recall that $\mathcal{S}_{\text{tr}} = \{(\boldsymbol{x}, y) : p_{\text{tr}}(\boldsymbol{x}, y) > 0\}$ and $\mathcal{S}_{\text{te}} = \{(\boldsymbol{x}, y) : p_{\text{te}}(\boldsymbol{x}, y) > 0\}$. Under case (i) or (ii), let us rewrite $R(\boldsymbol{f})$ and $J(\boldsymbol{f})$ with summations and integrals:

$$R(\boldsymbol{f}) = \sum_{y=1}^{C} \int_{\{\boldsymbol{x}:(\boldsymbol{x},y)\in\mathcal{S}_{\text{te}}\}} \ell(\boldsymbol{f}(\boldsymbol{x}), y) p_{\text{te}}(\boldsymbol{x}, y) \mathrm{d}\boldsymbol{x},$$

$$J(\boldsymbol{f}) = \sum_{y=1}^{C} \int_{\{\boldsymbol{x}:(\boldsymbol{x},y)\in\mathcal{S}_{\text{tr}}\}} \ell(\boldsymbol{f}(\boldsymbol{x}), y) w^*(\boldsymbol{x}, y) p_{\text{tr}}(\boldsymbol{x}, y) \mathrm{d}\boldsymbol{x}$$

$$= \sum_{y=1}^{C} \int_{\{\boldsymbol{x}:(\boldsymbol{x},y)\in\mathcal{S}_{\text{tr}}\}} \ell(\boldsymbol{f}(\boldsymbol{x}), y) p_{\text{te}}(\boldsymbol{x}, y) \mathrm{d}\boldsymbol{x},$$

where $w^*(\boldsymbol{x}, y) = p_{\text{te}}(\boldsymbol{x}, y)/p_{\text{tr}}(\boldsymbol{x}, y)$ is always well-defined over $\mathcal{S}_{\text{tr}}$ and we safely plugged this definition into the rewritten $J(\boldsymbol{f})$. Subsequently, in case (i), $\mathcal{S}_{\text{tr}} = \mathcal{S}_{\text{te}}$ and thus $J(\boldsymbol{f}) = R(\boldsymbol{f})$. In case (ii), $\mathcal{S}_{\text{tr}} \supset \mathcal{S}_{\text{te}}$ and then we further have

$$J(\boldsymbol{f}) = \sum_{y=1}^{C} \int_{\{\boldsymbol{x}:(\boldsymbol{x},y)\in\mathcal{S}_{\text{te}}\}} \ell(\boldsymbol{f}(\boldsymbol{x}), y) p_{\text{te}}(\boldsymbol{x}, y) \mathrm{d}\boldsymbol{x} + \int_{\{\boldsymbol{x}:(\boldsymbol{x},y)\in\mathcal{S}_{\text{tr}}\setminus\mathcal{S}_{\text{te}}\}} \ell(\boldsymbol{f}(\boldsymbol{x}), y) p_{\text{te}}(\boldsymbol{x}, y) \mathrm{d}\boldsymbol{x}.$$

By definition, $p_{\text{te}}(\boldsymbol{x}, y) = 0$ outside $\mathcal{S}_{\text{te}}$ including $\mathcal{S}_{\text{tr}} \setminus \mathcal{S}_{\text{te}}$, and thus $J(\boldsymbol{f}) = R(\boldsymbol{f})$. $\square$

**Theorem 2.** *In cases (iii) and (iv), IW is risk-inconsistent, and it holds that $J(\boldsymbol{f}) < R(\boldsymbol{f})$ for any $\boldsymbol{f}$.*
*Proof.* Since $w^*(\boldsymbol{x}, y)$ is well-defined over $\mathcal{S}_{\text{tr}}$ but it becomes ill-defined over $\mathcal{S}_{\text{te}} \setminus \mathcal{S}_{\text{tr}}$, we cannot naively replace the integral domain in $J(\boldsymbol{f})$ as in the proof of Theorem 1. In case (iii), $\mathcal{S}_{\text{tr}} \subset \mathcal{S}_{\text{te}}$, and consequently

$$R(\boldsymbol{f}) = \sum_{y=1}^{C} \int_{\{\boldsymbol{x}:(\boldsymbol{x},y)\in\mathcal{S}_{\text{tr}}\}} \ell(\boldsymbol{f}(\boldsymbol{x}), y) p_{\text{te}}(\boldsymbol{x}, y) \mathrm{d}\boldsymbol{x}$$

$$+ \sum_{y=1}^{C} \int_{\{\boldsymbol{x}:(\boldsymbol{x},y)\in\mathcal{S}_{\text{te}}\setminus\mathcal{S}_{\text{tr}}\}} \ell(\boldsymbol{f}(\boldsymbol{x}), y) p_{\text{te}}(\boldsymbol{x}, y) \mathrm{d}\boldsymbol{x}.$$

---

[5]For example, if $J(\boldsymbol{f}) = R(\boldsymbol{f})/2$ or if $J(\boldsymbol{f}) = R(\boldsymbol{f})^2$, minimizing $J(\boldsymbol{f})$ will give exactly the same optimal classifier as minimizing $R(\boldsymbol{f})$.

[6]In fact, if there is only the instance random variable $\boldsymbol{x}$ but no class-label random variable $y$, this result is simply the *importance-sampling identity* that can be found in many statistics textbooks. We prove it here to make our theoretical analyses self-contained and make later theoretical results easier to present/understand.

According to Theorem 1 for case (i), the first term in the rewritten $R(\boldsymbol{f})$ equals $J(\boldsymbol{f})$. Moreover, the second term is positive, since $\ell(\boldsymbol{f}(\boldsymbol{x}), y) > 0$ due to the positivity of $\ell$, and $p_{\text{te}}(\boldsymbol{x}, y) > 0$ over $\mathcal{S}_{\text{te}}$ including $\mathcal{S}_{\text{te}} \setminus \mathcal{S}_{\text{tr}}$. As a result, in case (iii), $R(\boldsymbol{f}) > J(\boldsymbol{f})$.

Similarly, in case (iv), we can split $\mathcal{S}_{\text{te}}$ into $\mathcal{S}_{\text{te}} \cap \mathcal{S}_{\text{tr}}$ and $\mathcal{S}_{\text{te}} \setminus \mathcal{S}_{\text{tr}}$ and decompose $R(\boldsymbol{f})$ as

$$R(\boldsymbol{f}) = \sum_{y=1}^{C} \int_{\{\boldsymbol{x}:(\boldsymbol{x},y)\in\mathcal{S}_{\text{te}}\cap\mathcal{S}_{\text{tr}}\}} \ell(\boldsymbol{f}(\boldsymbol{x}), y) p_{\text{te}}(\boldsymbol{x}, y) \mathrm{d}\boldsymbol{x}$$
$$+ \sum_{y=1}^{C} \int_{\{\boldsymbol{x}:(\boldsymbol{x},y)\in\mathcal{S}_{\text{te}}\setminus\mathcal{S}_{\text{tr}}\}} \ell(\boldsymbol{f}(\boldsymbol{x}), y) p_{\text{te}}(\boldsymbol{x}, y) \mathrm{d}\boldsymbol{x}.$$

Note that $\mathcal{S}_{\text{te}} \cap \mathcal{S}_{\text{tr}} \subset \mathcal{S}_{\text{tr}}$, so that according to Theorem 1 for case (ii), the first term equals $J(\boldsymbol{f})$. Following case (iii), the second term is positive. Therefore, in case (iv), $R(\boldsymbol{f}) > J(\boldsymbol{f})$. □

Theorem 1 implies that the objective of IW can be a good objective in cases (i) and (ii). Theorem 2 implies that the objective of IW may be a bad objective in cases (iii) and (iv). As a consequence, the theorems collectively address when and why IW methods can succeed/may fail.[7]

When the IW objective may be bad and IW methods may fail, whether an IW method fails or not depends on many factors, such as the underlying data distributions, the sampled data sets, the loss, the model, and the optimizer. To illustrate this phenomenon, here we give two concrete examples belonging to case (iii), where IW has no problem at all in one example and is as poor as random guessing in the other example.

**Two concrete examples**  We have seen the examples in Figure 2. In both examples, there are two classes marked with red and blue colors and distributed in four squares. Each square has a unit area and is the support of a uniform distribution of $\boldsymbol{x}$, i.e., $p(\boldsymbol{x}, 1) = 1$ and $p(\boldsymbol{x}, 0) = 0$ if its color is red, and $p(\boldsymbol{x}, 0) = 1$ and $p(\boldsymbol{x}, 1) = 0$ if its color is blue. There is a margin of 0.1 between two adjacent squares. The training distribution consists of the two squares on the left, and the test distribution consists of all the four squares. In the first example, on $\mathcal{S}_{\text{te}} \setminus \mathcal{S}_{\text{tr}}$, the label is red on the top and blue on the bottom, as same as the label on $\mathcal{S}_{\text{tr}}$. In the second example, on $\mathcal{S}_{\text{te}} \setminus \mathcal{S}_{\text{tr}}$, the label is blue on the top and red on the bottom, as opposite as the label on $\mathcal{S}_{\text{tr}}$.

We experimentally validated whether DIW works or not. The number of training data was 200. The number of validation data was only 4: we sampled one random point from each training square and added the center point of each test-only square. We can see that DIW performs very well in the first example, better than training from only the validation data; unfortunately, DIW performs very poorly in the second example, even worse than training from only the validation data.

The observed phenomenon should not be limited to DIW but be common to all IW methods. Here, we analyze why this phenomenon does not depend on the loss, the model, or the optimizer.

**Proposition 3.** *In the first example, IW is classifier-consistent, while in the second example, IW is classifier-inconsistent.*[8]
*Proof.* Without loss of generality, assume that $\ell$ is *classification-calibrated* (Bartlett et al., 2006).[9] Let $(c^{(1)}, c^{(2)})$ be the center of $\mathcal{S}_{\text{te}}$, and then the four squares are located on the top-left, bottom-left, top-right, and bottom-right of $(c^{(1)}, c^{(2)})$. For convenience, we abbreviate $f(x^{(1)}, x^{(2)})$ for $x^{(1)} > c^{(1)}, x^{(2)} > c^{(2)}$ as $f(+, +)$, $f(x^{(1)}, x^{(2)})$ for $x^{(1)} > c^{(1)}, x^{(2)} < c^{(2)}$ as $f(+, -)$, and so on.

Consider the first example. The minimizer of $R(f)$ can be any *Bayes-optimal classifier*, i.e., any $f$ such that $f(\cdot, +) > 0$ and $f(\cdot, -) < 0$. Next, on the top-left square, we have $p_{\text{te}}(\boldsymbol{x}, 1) = 1/4$, $p_{\text{te}}(\boldsymbol{x}, 0) = 0$, $p_{\text{tr}}(\boldsymbol{x}, 1) = 1/2$, $p_{\text{tr}}(\boldsymbol{x}, 0) = 0$, and thus $w^*(\boldsymbol{x}, y) = 1/2$. Likewise, on the bottom-left square, we have $w^*(\boldsymbol{x}, y) = 1/2$. As a result, $J(f) = \frac{1}{2}\mathbb{E}_{p_{\text{tr}}(\boldsymbol{x},y)}[\ell(\boldsymbol{f}(\boldsymbol{x}), y)]$, meaning that the minimizer of $J(f)$ can be any Bayes-optimal classifier on $\mathcal{S}_{\text{tr}}$, i.e., any $f$ such that $f(-, +) > 0$ and $f(-, -) < 0$. The simplest manner for $f$ to transfer its knowledge from $p_{\text{tr}}$ to $p_{\text{te}}$ is to have a linear decision boundary and extend it to $\mathcal{S}_{\text{te}} \setminus \mathcal{S}_{\text{tr}}$, so that $f(\cdot, +) > 0$ and $f(\cdot, -) < 0$ on $\mathcal{S}_{\text{te}}$. We can see that the set of minimizers is shared and thus IW is classifier-consistent.

---

[7] For any possible implementation, as long as it honestly implements the objective of IW, the quality of the objective will be reflected in the quality of the implementation.

[8] As the classifier $\boldsymbol{f}$ can only access the information on $\mathcal{S}_{\text{tr}}$ but must predict the class label on the whole $\mathcal{S}_{\text{te}}$, we assume that $\boldsymbol{f}$ would transfer its knowledge about $p_{\text{tr}}(\boldsymbol{x}, y)$ to $p_{\text{te}}(\boldsymbol{x}, y)$ in the simplest manner. Furthermore, we simplify $\boldsymbol{f}(\boldsymbol{x})$ into $f(x^{(1)}, x^{(2)})$, since it is binary classification where $\boldsymbol{x} \in \mathbb{R}^2$.

[9] The cross-entropy loss is *confidence-calibrated* and thus classification-calibrated (Sugiyama et al., 2022).

Consider the second example. The minimizer of $J(f)$ is still the same while the minimizer of $R(f)$ significantly changes to any $f$ such that $f(-,+) > 0$, $f(-,-) < 0$, $f(+,+) < 0$, and $f(+,-) > 0$. This non-linear decision boundary is a checkerboard where any two adjacent squares have opposite predictions. It is easy to see that IW is classifier-inconsistent and its test accuracy is 0.5. For binary classification with balanced classes, this accuracy is as poor as random guessing. $\square$

## 3 Generalized importance weighting (GIW)

We have seen two examples where IW is as good/bad as possible in case (iii). In practice, we cannot rely on the luck and hope that IW would work. In this section, we propose *generalized importance weighting* (GIW), which is still IW in cases (i) and (ii) and is better than IW in cases (iii) and (iv).

### 3.1 Expected objective of GIW

The key idea of GIW is to split the test support $\mathcal{S}_{\text{te}}$ into the *in-training* (IT) part $\mathcal{S}_{\text{te}} \cap \mathcal{S}_{\text{tr}}$ and the *out-of-training* (OOT) part $\mathcal{S}_{\text{te}} \setminus \mathcal{S}_{\text{tr}}$. More specifically, we introduce a third random variable, the support-splitting variable $s \in \{0, 1\}$, such that $s$ takes 1 on $\mathcal{S}_{\text{tr}}$ and 0 on $\mathcal{S}_{\text{te}} \setminus \mathcal{S}_{\text{tr}}$. As a result, the underlying joint density $p(\boldsymbol{x}, y, s)$ can be defined by $p_{\text{te}}(\boldsymbol{x}, y)$ as[10]

$$p(\boldsymbol{x}, y, s) = \begin{cases} p_{\text{te}}(\boldsymbol{x}, y) & \text{if } (\boldsymbol{x}, y) \in \mathcal{S}_{\text{tr}} \text{ and } s = 1, \text{ or } (\boldsymbol{x}, y) \in \mathcal{S}_{\text{te}} \setminus \mathcal{S}_{\text{tr}} \text{ and } s = 0, \\ 0 & \text{if } (\boldsymbol{x}, y) \in \mathcal{S}_{\text{tr}} \text{ and } s = 0, \text{ or } (\boldsymbol{x}, y) \in \mathcal{S}_{\text{te}} \setminus \mathcal{S}_{\text{tr}} \text{ and } s = 1. \end{cases} \quad (4)$$

Let $\alpha = p(s = 1)$. Then, the expected objective of GIW is defined as

$$J_{\text{G}}(\boldsymbol{f}) = \alpha \mathbb{E}_{p_{\text{tr}}(\boldsymbol{x}, y)}[w^*(\boldsymbol{x}, y)\ell(\boldsymbol{f}(\boldsymbol{x}), y)] + (1 - \alpha)\mathbb{E}_{p(\boldsymbol{x}, y|s=0)}[\ell(\boldsymbol{f}(\boldsymbol{x}), y)]. \quad (5)$$

The corresponding empirical version $\widehat{J}_{\text{G}}(\boldsymbol{f})$ will be derived in the next subsection. Before proceeding to the empirical objective of GIW, we establish risk consistency of GIW.

**Theorem 4.** *GIW is always risk-consistent for distribution shift problems.*
*Proof.* Let us work on the first term of $J_{\text{G}}(\boldsymbol{f})$ in (5). When $(\boldsymbol{x}, y) \in \mathcal{S}_{\text{tr}}$,

$$\alpha w^*(\boldsymbol{x}, y)p_{\text{tr}}(\boldsymbol{x}, y) = \alpha p_{\text{te}}(\boldsymbol{x}, y) = p(s = 1)p(\boldsymbol{x}, y \mid s = 1) = p(\boldsymbol{x}, y, s = 1),$$

where $p_{\text{te}}(\boldsymbol{x}, y) = p(\boldsymbol{x}, y \mid s = 1)$ given $(\boldsymbol{x}, y) \in \mathcal{S}_{\text{tr}}$ according to (4). Since $p(\boldsymbol{x}, y, s = 1) = 0$ on $\mathcal{S}_{\text{te}} \setminus \mathcal{S}_{\text{tr}}$, we have

$$\alpha \mathbb{E}_{p_{\text{tr}}(\boldsymbol{x}, y)}[w^*(\boldsymbol{x}, y)\ell(\boldsymbol{f}(\boldsymbol{x}), y)] = \sum_{y=1}^{C} \int_{\{\boldsymbol{x}:(\boldsymbol{x}, y) \in \mathcal{S}_{\text{te}}\}} \ell(\boldsymbol{f}(\boldsymbol{x}), y)p(\boldsymbol{x}, y, s = 1)\mathrm{d}\boldsymbol{x}. \quad (6)$$

Next, for the second term of $J_{\text{G}}(\boldsymbol{f})$, since $(1 - \alpha)p(\boldsymbol{x}, y \mid s = 0) = p(\boldsymbol{x}, y, s = 0)$, we have

$$(1 - \alpha)\mathbb{E}_{p(\boldsymbol{x}, y|s=0)}[\ell(\boldsymbol{f}(\boldsymbol{x}), y)] = \sum_{y=1}^{C} \int_{\{\boldsymbol{x}:(\boldsymbol{x}, y) \in \mathcal{S}_{\text{te}}\}} \ell(\boldsymbol{f}(\boldsymbol{x}), y)p(\boldsymbol{x}, y, s = 0)\mathrm{d}\boldsymbol{x}. \quad (7)$$

Note that $p(\boldsymbol{x}, y, s = 1) + p(\boldsymbol{x}, y, s = 0) = p_{\text{te}}(\boldsymbol{x}, y)$ according to (4). By adding (6) and (7), we can obtain that $J_{\text{G}}(\boldsymbol{f}) = R(\boldsymbol{f})$. This conclusion holds in all the four cases. $\square$

Theorem 4 is the main theorem of this paper. It implies that the objective of GIW can always be a good objective. Recall that IW is also risk-consistent in cases (i) and (ii), and it is interesting to see how IW and GIW are connected. By definition, given fixed $p_{\text{tr}}(\boldsymbol{x}, y)$ and $p_{\text{te}}(\boldsymbol{x}, y)$, if there exists a risk-consistent objective, it is unique. Indeed, in cases (i) and (ii), GIW is reduced to IW, simply due to that $\alpha = 1$ and $J_{\text{G}}(\boldsymbol{f}) = J(\boldsymbol{f})$ for any $\boldsymbol{f}$.

### 3.2 Empirical objective and practical implementation of GIW

Approximating $J_{\text{G}}(\boldsymbol{f})$ in (5) is more involved than approximating $J(\boldsymbol{f})$ in (2). Following (3), we need an empirical version $\widehat{w}(\boldsymbol{x}, y)$, and we need further to split the validation data $\mathcal{D}_{\text{v}}$ into two sets and estimate $\alpha$. Obviously, how to accurately split the validation data is the most challenging part. After splitting $\mathcal{D}_{\text{v}}$ and obtaining an estimate $\widehat{\alpha}$, the empirical objective will have two terms, where

---

[10]Here, $p(\boldsymbol{x}, y, s)$ accepts $(\boldsymbol{x}, y) \in \mathcal{S}_{\text{te}} \cup \mathcal{S}_{\text{tr}}$ rather than $(\boldsymbol{x}, y) \in \mathcal{S}_{\text{te}}$, and $s$ actually splits $\mathcal{S}_{\text{te}} \cup \mathcal{S}_{\text{tr}}$ into $\mathcal{S}_{\text{tr}}$ and $\mathcal{S}_{\text{te}} \setminus \mathcal{S}_{\text{tr}}$. This is exactly what we want, since $p_{\text{te}}(\boldsymbol{x}, y) = 0$ outside $\mathcal{S}_{\text{te}}$.

**Algorithm 1** Generalized importance weighting.

**Require:** model $\boldsymbol{f}_\theta$ parameterized by $\theta$;
training data set $\mathcal{D}_{\mathrm{tr}}$;
validation data set $\mathcal{D}_{\mathrm{v}}$;
batch sizes $m$, $n_1$, and $n_2$;
number of iterations $T$

1: **procedure 1.** VALDATASPLIT($\mathcal{D}_{\mathrm{tr}}, \mathcal{D}_{\mathrm{v}}$)
2:     `pretrain` $\boldsymbol{f}_\theta$ on $\mathcal{D}_{\mathrm{tr}}$
3:     `forward` the instances of $\mathcal{D}_{\mathrm{tr}}$ & $\mathcal{D}_{\mathrm{v}}$
4:     `retrieve` the transformed $Z_{\mathrm{tr}}$ & $Z_{\mathrm{v}}$
5:     `train` an O-SVM on $Z_{\mathrm{tr}}$ as $g(\boldsymbol{z})$
6:     `compute` $g(\boldsymbol{z})$ for every $\boldsymbol{z}$ in $Z_{\mathrm{v}}$
7:     `partition` $\mathcal{D}_{\mathrm{v}}$ into $\mathcal{D}_{\mathrm{v1}}$ & $\mathcal{D}_{\mathrm{v2}}$
8:     `estimate` $\hat{\alpha} = |\mathcal{D}_{\mathrm{v1}}|/|\mathcal{D}_{\mathrm{v}}|$
9:     **return** $\mathcal{D}_{\mathrm{v1}}, \mathcal{D}_{\mathrm{v2}}, \hat{\alpha}$
10: **end procedure**

1: **procedure 2.** MODELTRAIN($\mathcal{D}_{\mathrm{tr}}, \mathcal{D}_{\mathrm{v1}}, \mathcal{D}_{\mathrm{v2}}, \hat{\alpha}$)
2:     **for** $t = 1$ to $T$ **do**
3:         `sample` $S_{\mathrm{tr}}$ of size $m$ from $\mathcal{D}_{\mathrm{tr}}$
4:         `sample` $S_{\mathrm{v1}}$ of size $n_1$ from $\mathcal{D}_{\mathrm{v1}}$
5:         `sample` $S_{\mathrm{v2}}$ of size $n_2$ from $\mathcal{D}_{\mathrm{v2}}$
6:         `forward` the instances of $S_{\mathrm{tr}}$ & $S_{\mathrm{v1}}$
7:         `compute` the loss values as $L_{\mathrm{tr}}$ & $L_{\mathrm{v1}}$
8:         `match` the distributions of $L_{\mathrm{tr}}$ and $L_{\mathrm{v1}}$
            to obtain an estimated $\widehat{w}(\boldsymbol{x}, y)$
9:         `weight` $L_{\mathrm{tr}}$ with the estimated $\widehat{w}(\boldsymbol{x}, y)$
10:        `forward` the instances of $S_{\mathrm{v2}}$
11:        `backward` $\widehat{J}_{\mathrm{G}}(\boldsymbol{f}_\theta)$, and `update` $\theta$
12:    **end for**
13:    **return** the final $\boldsymbol{f}_\theta$
14: **end procedure**

the first term can be handled by any IW algorithm given training data and IT validation data, and the second term just involves OOT validation data.

To split $\mathcal{D}_{\mathrm{v}}$ and estimate $\alpha$, we employ the *one-class support vector machine* (O-SVM) (Schölkopf et al., 1999). Firstly, we pretrain a deep network for classification on the training data $\mathcal{D}_{\mathrm{tr}}$ a little bit and obtain a *feature extractor* from the pretrained deep network. Secondly, we apply the feature extractor on the instances in $\mathcal{D}_{\mathrm{tr}}$ and train an O-SVM based on the latent representation of these instances, giving us a score function $g(\boldsymbol{z})$ that could predict whether $p_{\mathrm{tr}}(\boldsymbol{x}) > 0$ or not, where $\boldsymbol{z}$ is the latent representation of $\boldsymbol{x}$. Thirdly, we apply the feature extractor on the instances in $\mathcal{D}_{\mathrm{v}}$ and then employ $g(\boldsymbol{z})$ to obtain the IT validation data $\mathcal{D}_{\mathrm{v1}} = \{(\boldsymbol{x}_i^{\mathrm{v1}}, y_i^{\mathrm{v1}})\}_{i=1}^{n_{\mathrm{v1}}}$ and the OOT validation data $\mathcal{D}_{\mathrm{v2}} = \{(\boldsymbol{x}_i^{\mathrm{v2}}, y_i^{\mathrm{v2}})\}_{i=1}^{n_{\mathrm{v2}}}$. Finally, $\alpha$ can be naturally estimated as $\widehat{\alpha} = n_{\mathrm{v1}}/n_{\mathrm{v}}$.

We have two comments on the split of $\mathcal{D}_{\mathrm{v}}$. The O-SVM $g(\boldsymbol{z})$ predicts whether $p_{\mathrm{tr}}(\boldsymbol{x}) > 0$ or not rather than whether $p_{\mathrm{tr}}(\boldsymbol{x}, y) > 0$ or not. This is because the $\boldsymbol{x}$-support change is often sufficiently informative in practice: when the $\boldsymbol{x}$-support changes, O-SVM can detect it; when the $(\boldsymbol{x}, y)$-support changes without changing the $\boldsymbol{x}$-support, it will be very difficult to train an O-SVM based on the loss-value representation of $(\boldsymbol{x}, y)$ to detect it, but such changes are very rare. The other comment is about the choice of the O-SVM. While there are more advanced one-class classification methods (Hido et al., 2011; Zaheer et al., 2020; Hu et al., 2020; Goldwasser et al., 2020) (see Perera et al. (2021) for a survey), the O-SVM is already good enough for the purpose (see Appendix C.1).

Subsequently, $\mathcal{D}_{\mathrm{v1}}$ can be viewed as being drawn from $p(\boldsymbol{x}, y \mid s = 1)$, and $\mathcal{D}_{\mathrm{v2}}$ can be viewed as being drawn from $p(\boldsymbol{x}, y \mid s = 0)$. Based on $\mathcal{D}_{\mathrm{tr}}$ and $\mathcal{D}_{\mathrm{v1}}$, we can obtain either $\widehat{w}(\boldsymbol{x}, y)$ or $\widehat{w}_i$ for each $(\boldsymbol{x}_i^{\mathrm{tr}}, y_i^{\mathrm{tr}})$ by IW. IW has no problem here since the split of $\mathcal{D}_{\mathrm{v}}$ can reduce case (iii) to case (i) and case (iv) to case (ii). In the implementation, we employ DIW (Fang et al., 2020) because it is friendly to deep learning and it is a state-of-the-art IW method. Finally, the empirical objective of GIW can be expressed as

$$\widehat{J}_{\mathrm{G}}(\boldsymbol{f}) = \tfrac{n_{\mathrm{v1}}}{n_{\mathrm{v}} n_{\mathrm{tr}}} \sum_{i=1}^{n_{\mathrm{tr}}} \widehat{w}(\boldsymbol{x}_i^{\mathrm{tr}}, y_i^{\mathrm{tr}}) \ell(\boldsymbol{f}(\boldsymbol{x}_i^{\mathrm{tr}}), y_i^{\mathrm{tr}}) + \tfrac{1}{n_{\mathrm{v}}} \sum_{j=1}^{n_{\mathrm{v2}}} \ell(\boldsymbol{f}(\boldsymbol{x}_j^{\mathrm{v2}}), y_j^{\mathrm{v2}}), \tag{8}$$

where the two expectations in $J_{\mathrm{G}}(\boldsymbol{f})$ are approximated separately with $\mathcal{D}_{\mathrm{tr}}$ and $\mathcal{D}_{\mathrm{v2}}$.[11]

The practical implementation of GIW is presented in Algorithm 1. Here, we adopt the hidden-layer-output representation for O-SVM in VALDATASPLIT and the loss-value representation for DIW in

---

[11]In GIW, although $J_{\mathrm{G}}(\boldsymbol{f}) = R(\boldsymbol{f})$, $\widehat{J}_{\mathrm{G}}(\boldsymbol{f})$ is not an unbiased estimator of $J_{\mathrm{G}}(\boldsymbol{f})$, exactly the same as what happened in IW. Nevertheless, $\widehat{J}_{\mathrm{G}}(\boldsymbol{f})$ can still be statistically consistent with $J_{\mathrm{G}}(\boldsymbol{f})$ under mild conditions if $\widehat{w}(\boldsymbol{x}, y)$ is consistent with $w^*(\boldsymbol{x}, y)$. Specifically, though the outer weighted classification is an optimization problem, the inner weight estimation is an estimation problem. The statistical consistency of $\widehat{w}(\boldsymbol{x}, y)$ requires *zero approximation error* and thus *non-parametric estimation* is preferred. For kernel-based IW methods such as Huang et al. (2007) and Kanamori et al. (2009), it holds that as $n_{\mathrm{tr}}, n_{\mathrm{v}} \to \infty$, $\widehat{w}(\boldsymbol{x}, y) \to w^*(\boldsymbol{x}, y)$ under mild conditions. If so, we can establish statistical consistency of $\widehat{J}_{\mathrm{G}}(\boldsymbol{f})$ with $J_{\mathrm{G}}(\boldsymbol{f})$. If we further assume that the function class of $\boldsymbol{f}$ has a bounded complexity and $\ell$ is bounded and Lipschitz continuous, we can establish statistical consistency of $R(\widehat{\boldsymbol{f}})$ with $R(\boldsymbol{f}^*)$, where $\widehat{\boldsymbol{f}}(\boldsymbol{x})$ and $\boldsymbol{f}^*(\boldsymbol{x})$ are the minimizers of $\widehat{J}_{\mathrm{G}}(\boldsymbol{f})$ and $R(\boldsymbol{f})$.

Table 1: Specification of benchmark datasets, tasks, distribution shifts, and models.

| Dataset | Task | Training data | Test data | Model |
|---------|------|---------------|-----------|-------|
| MNIST | odd and even digits | 4 digits (0-3) | 10 digits (0-9)[*] | LeNet-5 |
| Color-MNIST | 10 digits | digits in red | digits in red/blue/green | LeNet-5 |
| CIFAR-20 | 20 superclasses | 2 classes per superclass | 5 classes per superclass | ResNet-18 |

See LeCun et al. (1998) for MNIST and Krizhevsky and Hinton (2009) for CIFAR-20. Color-MNIST is modified from MNIST. The model is a modified LeNet-5 (LeCun et al., 1998) or ResNet-18 (He et al., 2016). Please find in Appendix B.1 the details. *All setups in the table are for case (iii); for MNIST in case (iv), the test data consist of 8 digits (2-9).

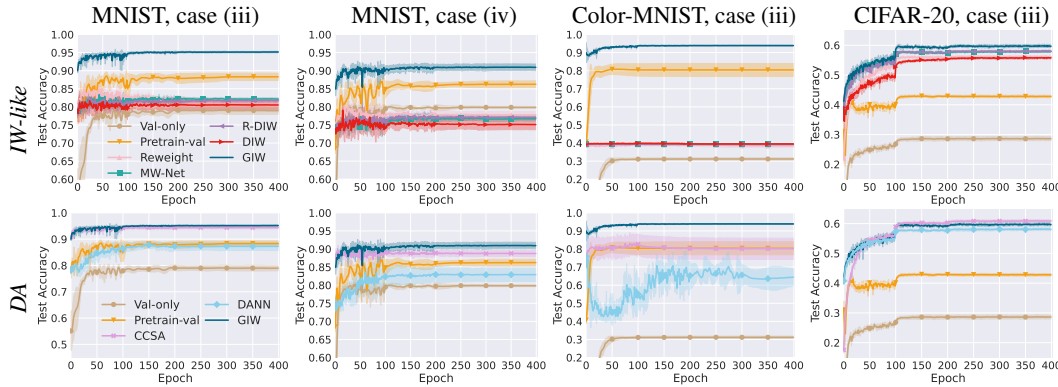

Figure 3: Comparisons with IW-like and DA baselines under support shift (5 trails).

MODELTRAIN. This algorithm design is convenient for both O-SVM and DIW; the hidden-layer-output representation for DIW has been tried and can be found in Section 4.3.

# 4 Experiments

In this section, we empirically evaluate GIW and compare it with baseline methods.[12] To see how effective it is in cases (iii) and (iv), we designed two distribution shift (DS) patterns. In the first pattern, DS comes solely from the mismatch between the training and test supports, and we call it support shift (SS). Under SS, it holds that $w^*(\boldsymbol{x}, y)$ equals $\alpha$ in case (iii) and 0 or another constant in case (iv), simply due to renormalization after imposing SS. Hence, the challenge is to accurately split $\mathcal{D}_v$. In the second pattern, there is some genuine DS (e.g., label noise or class-prior shift) on top of SS, and we call it support-distribution shift. Since $w^*(\boldsymbol{x}, y)$ is no longer a constant, we face the challenge to accurately estimate $w^*(\boldsymbol{x}, y)$. Additionally, we conducted an ablation study to better understand the behavior of GIW. Detailed setups and more results are given in Appendices B & C.

The baseline methods involved in our experiments are as follows.
- *Val-only*: using only $\mathcal{D}_v$ to train the model from scratch.
- *Pretrain-val*: first pretraining on $\mathcal{D}_{tr}$ and then training on $\mathcal{D}_v$.
- *Reweight*: learning to reweight examples (Ren et al., 2018).
- *MW-Net*: meta-weight-net (Shu et al., 2019), a parametric version of Reweight.
- *DIW*: dynamic importance weighting (Fang et al., 2020).
- *R-DIW*: DIW where IW is done with *relative density-ratio estimation* (Yamada et al., 2011).
- *CCSA*: classification and contrastive semantic alignment (Motiian et al., 2017).
- *DANN*: domain-adversarial neural network (Ganin et al., 2016).

## 4.1 Experiments under support shift

We first conducted experiments under support shift on benchmark datasets. The setups are summarized in Table 1. For MNIST, our task was to classify odd and even digits, where the training set has only 4 digits (0-3), while the test set has 10 digits (0-9) in case (iii) and 8 digits (2-9) in case (iv). For Color-MNIST, our task was to classify 10 digits; the dataset was modified from MNIST in such a way that the digits in the training set are colored in red while the digits in the test/validation set are colored

---

[12]Our implementation of GIW is available at https://github.com/TongtongFANG/GIW.



| Val-only | DIW | Pretrain-val | GIW |

Figure 4: Visualizations of the learned convolution kernels on Color-MNIST under support shift.

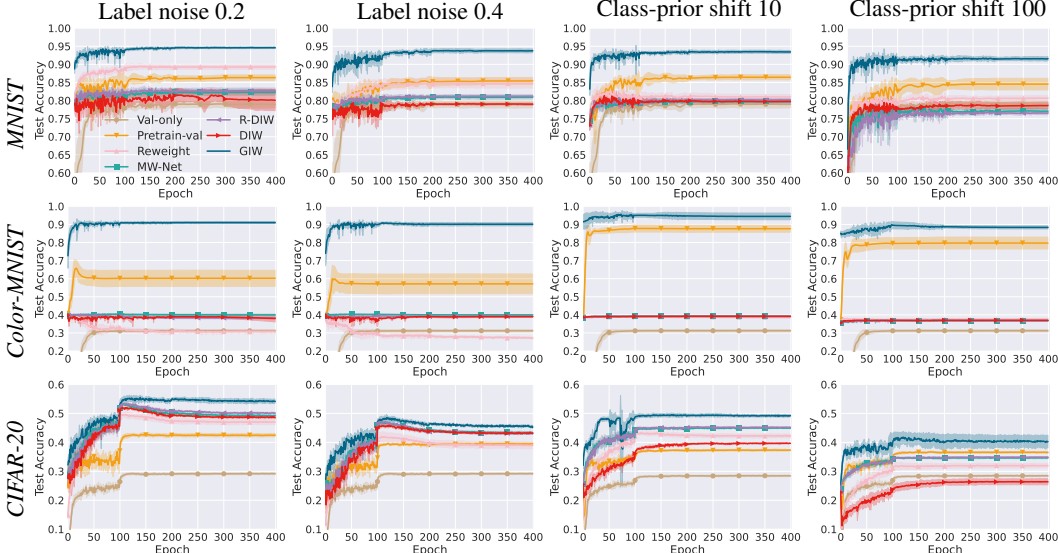

Figure 5: Comparisons with IW-like baselines under support-distribution shift (5 trails).

in red/green/blue evenly. For CIFAR-100, our task was to classify the 20 predefined superclasses and thus we call it CIFAR-20; the training set contains data from 2 out of the 5 classes for each superclass while the test set contains all classes. For validation set, we sampled 2 data points per test digit for MNIST and Color-MNIST, and 10 data points per class for CIFAR-100.

Figure 3 shows the results on MNIST, Color-MNIST, and CIFAR-20 under support shift[13], where GIW generally outperforms IW-like and domain adaptation (DA) baselines. We also confirmed that $\alpha$ in (5) is accurately estimated in Appendix C.1. To further investigate how GIW works, we visualized the learned convolution kernels (i.e., weights) for Color-MNIST experiments in Figure 4, where the more observed color represents the larger weights learned on that color channel. Only GIW recovers the weights of all color channels while learning useful features, however, other methods fail to do so.

## 4.2 Experiments under support-distribution shift

We further imposed additional distribution shift, i.e., adding label noise or class-prior shift, on top of the support shift following the same setup in Table 1. Here we only show the results in case (iii) and defer the results in case (iv) to Appendix C.4 due to the space limitation.

**Label-noise experiments** In addition to the support shift, we imposed label noise by randomly flipping a label to other classes with an equal probability, i.e., the noise rate. The noise rates are set as $\{0.2, 0.4\}$ and the corresponding experimental results are shown in Figure 5 and 6. We can see that compared with baselines, GIW performs better and tends to be robust to noisy labels.

**Class-prior-shift experiments** On top of the support shift, we imposed class-prior shift by reducing the number of training data in half of the classes to make them minority classes (as opposed to majority classes). The sample size ratio per class between the majority and minority classes is defined as $\rho$, chosen from $\{10, 100\}$. To fully use the data from the minority class, we did not split the validation data in class-prior-shift experiments and used all validation data in optimizing the two

---

[13]It is difficult to have the same number of mini-batches per epoch for the training and validation data. To this end, we adopt the definition of an epoch as a single loop over the training data, not the validation data.

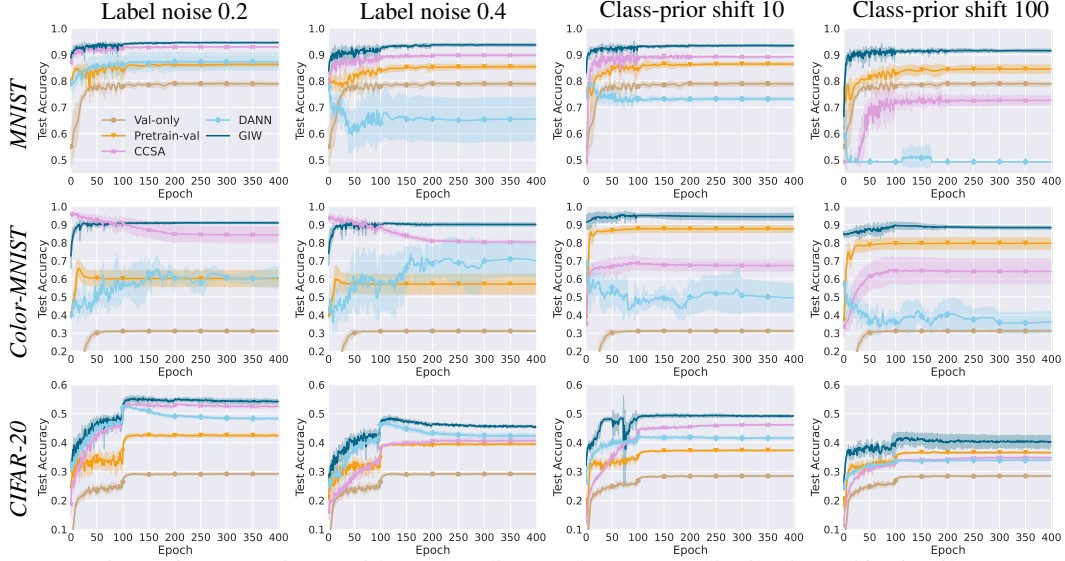

Figure 6: Comparisons with DA baselines under support-distribution shift (5 trails).

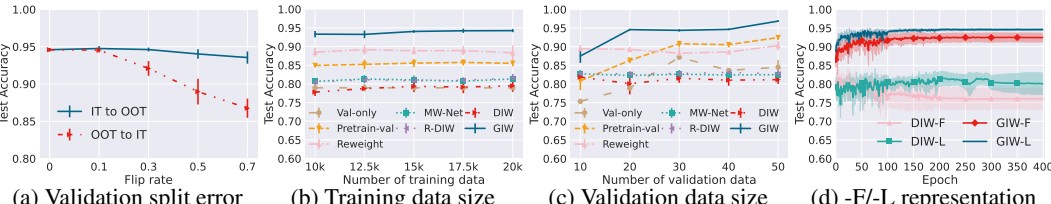

(a) Validation split error    (b) Training data size    (c) Validation data size    (d) -F/-L representation

Figure 7: Impact of different validation data splits, sample sizes, and data representations.

terms in (8). In Figure 5 and 6, we can see that GIW consistently performs better than the baselines. Note that though domain adaptation (DA) baselines (i.e., CCSA & DANN) may achieve comparable performance to GIW under the support shift in Figure 3, their effectiveness declines significantly when confronting additional distribution shifts (e.g., label noise or class-prior shift) in Figure 6.

## 4.3 Ablation study

Finally, we performed an ablation study on MNIST under 0.2 label noise. Figure 7(a) and Table 2 in Appendix C.2 present the negative impact of validation data split errors by randomly flipping the IT/OOT data into the OOT/IT part. We can see that GIW is reasonably robust to split errors, and flipping OOT to IT is more problematic than the other direction since it makes the empirical risk estimator of GIW more similar to that of the standard IW. In Figure 7(b), the performance of all methods remains consistent across different values of $n_{tr} \in \{10000, 12500, 15000, 17500, 20000\}$. In Figure 7(c), GIW performs better with more validation data, e.g., when $n_v$ increases from 10 to 20. Moreover, we compare the loss-value (-L) with the hidden-layer-output (-F) representation of data used in the DIW and GIW methods. Figure 7(d) shows GIW-L outperforms others.

## 5 Conclusions

We characterized distribution shift into four cases according to support shift to gain a deeper understanding of IW. Consequently, we found that IW is provably good in two cases but provably poor in the other two cases. Then, we proposed GIW which is a strict generalization of IW and is provably favorable in all the four cases. GIW is safer to be used in practice as it is difficult to know in which case the problem to be solved is. That said, there are still some potential limitations and thus future directions to work on: GIW requires *exactly test-distributed* validation data, which is restrictive; it requires *not very small* validation data, which is demanding; a small amount of OOT validation data *joins training directly* and is used for validation simultaneously, which might lead to imperceptible overfitting as well as overoptimism about the OOT part of the test support.

## Acknowledgments and Disclosure of Funding

TF was supported by JSPS KAKENHI Grant Number 23KJ0438 and the Institute for AI and Beyond, UTokyo. NL was funded by the Deutsche Forschungsgemeinschaft (DFG, German Research Foundation) under Germany's Excellence Strategy – EXC number 2064/1 – Project number 390727645. MS was supported by the Institute for AI and Beyond, UTokyo.

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

# Supplementary Material

## A   Related work

In this section, we discuss relevant prior studies for addressing distribution shift problems, including importance weighting (IW), IW-like methods, and domain adaptation (DA).

**Importance weighting (IW)**   IW has been a powerful tool for mitigating the influence of distribution shifts. The general idea is first to estimate the importance, which is the test over training density ratio, and then train a classifier by weighting the training losses according to the importance. Numerous IW methods have been developed in this manner, utilizing different techniques for importance estimation.

The *kernel mean matching* (KMM) approach (Huang et al., 2007) learns the importance function by matching the distributions of training and test data in terms of the maximum mean discrepancy in a reproducing kernel Hilbert space, while the *Kullback-Leibler importance estimation procedure* (KLIEP) (Sugiyama et al., 2007b) employs the KL divergence for density fitting and the *least-squares importance fitting* (LSIF) (Kanamori et al., 2009) employs squared loss for importance fitting. The *unconstrained LSIF* (uLSIF) (Kanamori et al., 2009) is an approximation version of LSIF that removes the non-negativity constraint in optimization, allowing for more efficient computation. To boost the performance of such traditional IW methods, *dynamic importance weighting* (DIW) (Fang et al., 2020) is recently proposed to make them compatible with stochastic optimizers, thereby facilitating their effective integration with deep learning frameworks.

However, in order to establish a well-defined notion of importance, all IW methods including DIW assume cases (i) and (ii), while they become problematic in cases (iii) and (iv).

**IW-like methods**   A relevant IW invariant is the *relative unconstrained least-squares importance fitting* (RuLSIF) (Yamada et al., 2011), which considers a smoothed and bounded extension of the importance. Instead of estimating the importance $w^*(\boldsymbol{x}, y) = p_{\text{te}}(\boldsymbol{x}, y)/p_{\text{tr}}(\boldsymbol{x}, y)$, they estimate the $\eta$-relative importance $w_\eta^*(\boldsymbol{x}, y) = p_{\text{te}}(\boldsymbol{x}, y)/\left(\eta p_{\text{te}}(\boldsymbol{x}, y) + (1 - \eta)p_{\text{tr}}(\boldsymbol{x}, y)\right)$ where $0 \leq \eta \leq 1$. While the relative importance is well-defined in cases (iii) and (iv), experiments have demonstrated that it is inferior to GIW, since its training does not incorporate any out-of-training (OOT) data.

Moreover, some reweighting approaches based on bilevel optimization look like DIW in the sense of iterative training between weighted classification on the training data for learning the classifier and weight estimation with the help of a small set of validation data for learning the weights (Jiang et al., 2018; Ren et al., 2018; Shu et al., 2019). However, they encounter a similar issue as IW and RuLSIF, where validation data is solely used for learning the weights, while the training data (without any OOT data) is used for training the classifier. This makes them hard to handle the cases (iii) and (iv).

**Domain adaptation (DA)**   DA relates to DS problems where the $p_{\text{te}}(\boldsymbol{x}, y)$ and the $p_{\text{tr}}(\boldsymbol{x}, y)$ are usually named as target and source domain distributions (Ben-David et al., 2006), or in-domain and out-of-domain distributions (Duchi et al., 2016). It can be categorized into *supervised* DA (SDA) or *unsupervised* DA (UDA): the former has labeled test data while the latter has unlabeled test data. The setting of SDA is similar as that of GIW. One representative SDA work is *classification and contrastive semantic alignment* (CCSA) (Motiian et al., 2017) method. In CCSA, a contrastive semantic alignment loss is added to the classification loss, for minimizing the distances between the samples that come from the same class and maximizing the distances between samples from different classes. In DA research, UDA is more popular than SDA. Based on different assumptions, UDA involves learning domain-invariant (Ganin et al., 2016) or conditional domain-invariant features (Gong et al., 2016), or giving pseudo labels to the target domain data (Saito et al., 2017).

Note that DA can refer to either certain problem settings or the corresponding learning methods (or both). When regarding it as problem settings, SDA is exactly the same as joint shift and UDA is fairly similar to covariate shift, which assumes $p(y|\boldsymbol{x})$ doesn't change too much between the training and test domain. When regarding it as learning methods, the philosophy of both SDA and UDA is to find good representations to link the source and target domains and transfer knowledge from the source domain to the target domain, which is totally different from the philosophy of IW.

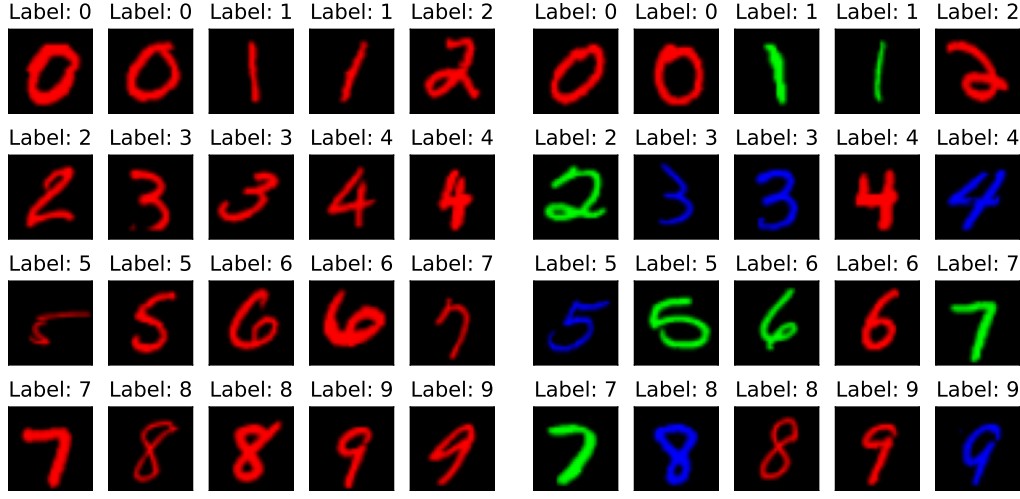

Training data          Validation data

Figure 8: A plot of the training data and validation data in Color-MNIST dataset.

## B   Supplementary information on experimental setup

In this section, we present supplementary information on the experimental setup. All experiments were implemented using PyTorch 1.13.1[14] and carried out on NVIDIA Tesla V100 GPUs[15].

### B.1   Datasets and base models

**MNIST**   MNIST (LeCun et al., 1998) is a 28*28 grayscale image dataset for 10 hand-written digits (0–9). The original dataset includes 60,000 training data and 10,000 test data. See `http://yann.lecun.com/exdb/mnist/` for more details.

In the experiments, we converted it for binary classification to classify even/odd digits as follows:
- Class 0: digits '0', '2', '4', '6', and '8';
- Class 1: digits '1', '3', '5', '7', and '9'.

In our setup, the training data only included 4 digits (0-4). The test data could access all digits (0-9) in case (iii) and 8 digits (2-9) in case (iv). Since the number of training data was reduced, we added two data augmentations to the training and validation data: random rotation with degree 10 and random affine transformation with degree 10, translate of (0.1, 0.1) and scale of (0.9, 1.1). Note that the data augmentations were only added during procedure 2 MODELTRAIN in Algorithm 1.

Accordingly, we modified LeNet-5 (LeCun et al., 1998) as the base model for MNIST:

     0th (input) layer:  (32*32)-
           1st layer:  C(5*5,6)-S(2*2)-
         2nd layer:  C(5*5,16)-S(2*2)-
         3rd layer:  FC(120)-
   4th to 5th layer:  FC(84)-2,

where C(5*5,6) represents a 5*5 convolutional layer with 6 output channels followed by ReLU, S(2*2) represents a max-pooling layer with a filter of size 2*2, and FC(120) represents a fully connected layer with 120 outputs followed by ReLU, etc. The hidden-layer-output representation of data used in the implementation was the normalized output extracted from the 3rd layer.

**Color-MNIST**   Color-MNIST was modified from MNIST for 10 hand-written digit classification, where the digits in training data were colored in red and the digits in test/validation data were colored in either red, green or blue evenly. See Figure 8 for a plot of the training data and validation data. We did not add any data augmentation for experiments on Color-MNIST.

---

[14] `https://pytorch.org`
[15] `https://www.nvidia.com/en-us/data-center/v100/`

To process RGB input data, we modified LeNet-5 as the base model for Color-MNIST:

         0th (input) layer:  (32*32*3)-
                 1st layer:  C(5*5,20)-S(2*2)-
                2nd layer:  C(5*5,50)-S(2*2)-
                 3rd layer:  FC(120)-
         4th to 5th layer:  FC(84)-10,

where the abbreviations and the way of extracting the hidden-layer-output representation of data were the same as that in MNIST.

**CIFAR-20**   CIFAR-100 (Krizhevsky and Hinton, 2009) is a 32*32 colored image dataset in 100 classes, grouped in 20 superclasses. It contains 50,000 training data and 10,000 test data. We call this dataset CIFAR-20 since we use it for 20-superclass classification—the predefined superclasses and classes as shown below, where each superclass includes five distinct classes. See `https://www.cs.toronto.edu/~kriz/cifar.html` for more details.

| **Superclass** | **Class** |
| --- | --- |
| aquatic mammals | (beaver, dolphin), otter, seal, whale |
| fish | (aquarium fish, flatfish), ray, shark, trout |
| flowers | (orchids, poppies), roses, sunflowers, tulips |
| food containers | (bottles, bowls), cans, cups, plates |
| fruit and vegetables | (apples, mushrooms), oranges, pears, sweet peppers |
| household electrical devices | (clock, computer keyboard), lamp, telephone, television |
| household furniture | (bed, chair), couch, table, wardrobe |
| insects | (bee, beetle), butterfly, caterpillar, cockroach |
| large carnivores | (bear, leopard), lion, tiger, wolf |
| large man-made outdoor things | (bridge, castle), house, road, skyscraper |
| large natural outdoor scenes | (cloud, forest), mountain, plain, sea |
| large omnivores and herbivores | (camel, cattle), chimpanzee, elephant, kangaroo |
| medium-sized mammals | (fox, porcupine), possum, raccoon, skunk |
| non-insect invertebrates | (crab, lobster), snail, spider, worm |
| people | (baby, boy), girl, man, woman |
| reptiles | (crocodile, dinosaur), lizard, snake, turtle |
| small mammals | (hamster, mouse), rabbit, shrew, squirrel |
| trees | (maple, oak), palm, pine, willow |
| vehicles 1 | (bicycle, bus), motorcycle, pickup truck, train |
| vehicles 2 | (lawn-mower, rocket), streetcar, tank, tractor |

In our setup, the training data only included the data in 2 out of the 5 classes per superclass, i.e., the classes in ( ) were seen by the training data as shown above. The test data included the data in all classes. Since the number of training data was reduced, we added several data augmentations to the training and validation data: random horizontal flip, random vertical flip, random rotation of degree 10 and random crop of size 32 with padding 4. Same as that in MNIST experiments, the data augmentations were only added during procedure 2 MODELTRAIN in Algorithm 1.

As for the base model for CIFAR-20, we adopted ResNet-18 (He et al., 2016) as follows:

          0th (input) layer:  (32*32*3)-
         1st to 5th layers:  C(3*3, 64)-[C(3*3, 64), C(3*3, 64)]*2-
         6th to 9th layers:  [C(3*3, 128), C(3*3, 128)]*2-
       10th to 13th layers:  [C(3*3, 256), C(3*3, 256)]*2-
       14th to 17th layers:  [C(3*3, 512), C(3*3, 512)]*2-
                18th layer:  Global Average Pooling-20,

where [ ·, · ] denotes a building block (He et al., 2016) and [·]*2 means 2 such blocks, etc. Batch normalization (Ioffe and Szegedy, 2015) was applied after convolutional layers. The hidden-layer-output representation of data was the normalized output after pooling operation in the 18th layer.

## B.2   Experiments under support shift

For all compared methods except Val-only, we pre-trained the model for 10 epochs as the initialization. For the one-class support vector machine (O-SVM) (Schölkopf et al., 1999), we adopted

the implementation from scikit-learn[16], where the radial basis function (RBF) kernel was used: $k(\boldsymbol{x}_i, \boldsymbol{x}_j) = e^{-\gamma \|\boldsymbol{x}_i - \boldsymbol{x}_j\|^2}$ with $\gamma = 10000$. All other hyperparameters about O-SVM were set as the default. For the distribution matching by dynamic importance weighting (DIW) (Fang et al., 2020), we again used the RBF kernel where $\gamma$ was the median distance between the training data. And we used $\boldsymbol{K} + \omega I$ as the kernel matrix $\boldsymbol{K}$, where $I$ was an identity matrix and $\omega$ was set to be 1e-05. The upper bound of weights was set as 50.

In all experiments under support shift, Adam (Kingma and Ba, 2015) was used as the optimizer, the learning rate was 0.0005, decaying every 100 epochs by multiplying a factor of 0.1, and the batch size was set as 256. For MNIST, Color-MNIST, and CIFAR-20 experiments, the weight decay was set as 0.005, 0.002, and 0.0001, respectively.

### B.3 Experiments under support-distribution shift

For experiments under support-distribution shift, all the setups and hyperparameters about the initialization, O-SVM, and distribution matching by DIW were the same as that in Section B.2. Moreover, the same that Adam was used as the optimizer, the learning rate was 0.0005, decaying every 100 epochs by multiplying a factor of 0.1, and the batch size was set as 256. Next we show the setups and hyperparameters specific to the support-distribution shift.

**Label-noise experiments** On top of the support shift in Section B.2, we added a symmetric label noise to the training data, where a label may flip to all other classes with an equal probability (this probability was defined as the noise rate, set as $\{0.2, 0.4\}$). The type of label noise and the noise rate were unknown to the model. For MNIST, Color-MNIST, and CIFAR-20 experiments, the weight decay was set as 0.005, 0.002, and 0.008, respectively.

**Class-prior shift experiments** We induced class-prior shift in the training data by randomly sampling half of the classes as minority classes (other classes were the majority classes) and reducing the number of samples in minority classes. The sample size ratio per class between the majority and minority classes was $\rho$, chosen from $\{10, 100\}$. The randomly selected minority classes in class-prior shift experiments were shown as follows:

- MNIST: class of odd digits;
- Color-MNIST: digits '1', '2', '6', '7', and '8';
- CIFAR-20: superclasses of 'fish', 'fruit and vegetables', 'household electrical device', 'household furniture', 'large carnivores', 'large omnivores and herbivores', 'medium-sized mammals', 'people', 'small mammals', and 'vehicles 1'.

For MNIST, Color-MNIST, and CIFAR-20 experiments, the weight decay was set as 0.005, 1e-05, and 1e-07, respectively. Since we did not split the validation data in class-prior shift experiments, we set the $\alpha$ in (5) as 0.5 for class-prior shift experiments on all datasets.

## C Supplementary experimental results

In this section, we present supplementary experimental results, including the histogram plots of the learned O-SVM score, more ablation study results on the validation data split error, visualizations of convolution kernels for all methods under label noise, additional experimental results for case (iv), and the summary of classification accuracy.

### C.1 On the learned O-SVM score

Figure 9 shows the histogram plots of the learned O-SVM score on MNIST, Color-MNIST, and CIFAR-20 under support shift. From the histogram plots, we observe that the score distribution consists of two peaks without overlapping; therefore, any value between the two peaks (e.g., 0.4) could be made as a threshold to split the validation data into two parts. If the score of validation data is higher than the threshold, then the data is identified as an (in-training) IT validation data; otherwise, it is an (out-of-training) OOT validation data.

---

[16] https://scikit-learn.org/stable/modules/generated/sklearn.svm.OneClassSVM.html

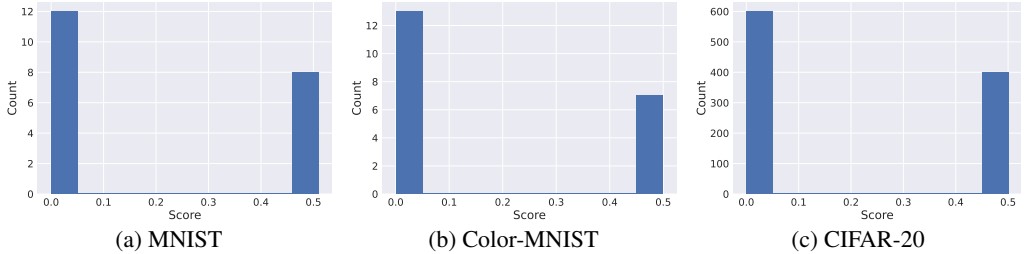

|       (a) MNIST       |    (b) Color-MNIST    |      (c) CIFAR-20      |

Figure 9: Histogram plots of the learned O-SVM score under support shift.

Table 2: Mean accuracy (standard deviation) in percentage on MNIST over the last ten epochs under support shift with label noise (5 trials). OOT/IT is short for out-of-training/in-training data. OOT → IT means OOT data flips to IT data and vice versa. LN is short for label noise. Flip rate is the percentage of OOT/IT data that randomly flipped to IT/OOT data.

| Flip rate | OOT → IT | | IT → OOT | |
|---|---|---|---|---|
| | LN 0.2 | LN 0.4 | LN 0.2 | LN 0.4 |
| 0 | 94.60 (0.22) | 93.74 (0.53) | 94.60 (0.22) | 93.74 (0.53) |
| 0.1 | 94.54 (0.31) | 94.04 (0.54) | 94.76 (0.33) | 93.66 (0.44) |
| 0.3 | 92.09 (0.99) | 92.38 (1.14) | 94.62 (0.26) | 92.78 (0.90) |
| 0.5 | 88.98 (1.73) | 88.83 (2.26) | 94.03 (0.64) | 92.25 (1.36) |
| 0.7 | 86.76 (1.28) | 85.94 (2.14) | 93.55 (0.82) | 90.53 (1.22) |

After splitting the validation data, $\alpha$ in (5) is estimated as the ratio of the sample size between the IT validation data and the whole validation data, i.e., $\widehat{\alpha} = n_{v1}/n_v$. For example, in Figure 9(a), $\widehat{\alpha} = \frac{8}{8+12} = 0.4$, which is equal to the true value in MNIST experiments. Similarly, it can be verified that the $\alpha$ in Figure 9(b) and 9(c) are also accurately estimated.

### C.2   Additional results on the validation data split error

As mentioned in the main paper in Figure 7(a), generally GIW is quite robust to split errors and OOT→IT is more problematic than the other direction. Here we present additional results on MNIST under 0.4 label noise in Table 2. We can further observe that as the label noise increases from 0.2 to 0.4, IT→OOT is more susceptible to such negative impacts than the other direction.

### C.3   Visualizations of convolution kernels under label noise

Then, we visualized the learned convolution kernels for all methods on Color-MNIST under 0.2 label noise. From Figure 10, we can see that the results aligned with the discussions about Figure 4. Previous IW or IW-like methods (i.e., DIW, R-DIW, Reweight and MW-Net), DA methods (i.e., CCSA and DANN), and Pretrain-val learned most weights on the red channel, which may cause the failure on the test data with green/blue color. Although Val-only had weights on all color channels, it may fail to learn useful data representation due to its limited training data. Only GIW could successfully recover the weights on all color channels while capturing the data representation.

### C.4   Additional experimental results for case (iv)

Here we present the results on MNIST in case (iv) under distribution-support shift, comparing with IW-like and domain adaptation (DA) baselines. From Figure 11, we can see that GIW outperforms other methods by a large margin in case (iv) under both label-noise and class-prior-shift settings.

### C.5   Summary of classification accuracy

Table 3 and 4 present the mean accuracy (standard deviation) in percentage on MNIST, Color-MNIST, and CIFAR-20 over the last ten epochs under support shift, comparing with IW-like methods and domain adaptation (DA) methods respectively, corresponding to Figure 3. Table 5 and 6 present such results under support-distribution shift, corresponding to Figure 5 and 6. Table 7 and 8 shows the summary of results in case (iv) on MNIST, corresponding to Figure 11.

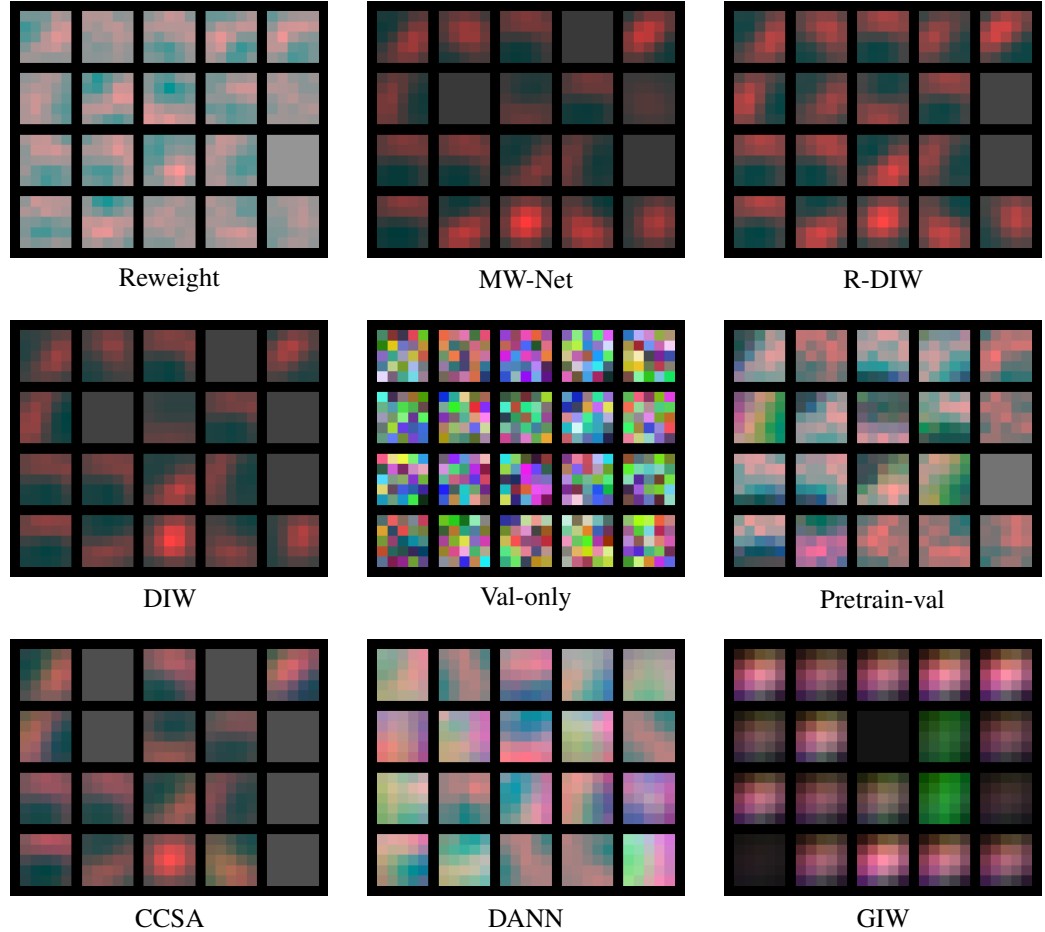

Figure 10: Visualizations of the learned convolution kernels on Color-MNIST under 0.2 label noise.

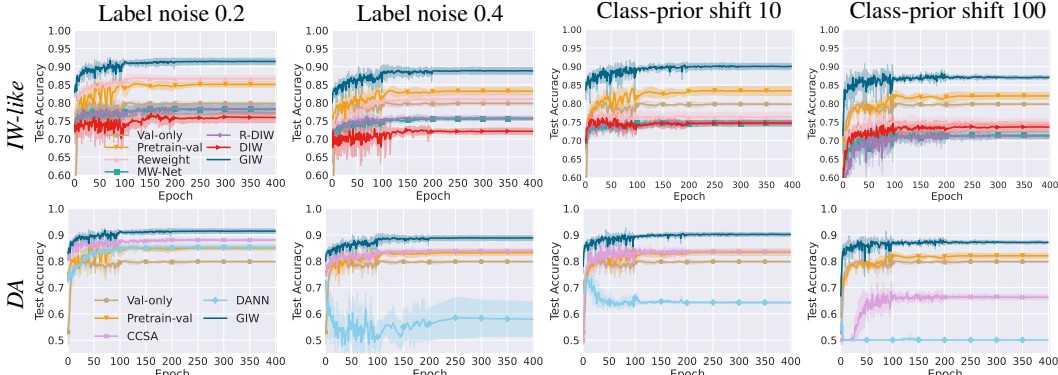

Figure 11: Results on MNIST in case (iv) under distribution-support shift (5 trails).

Table 3: Mean accuracy (standard deviation) in percentage on MNIST, Color-MNIST, and CIFAR-20 over the last ten epochs under support shift in case (iii) with IW-like baselines (5 trials). Best and comparable methods (paired *t*-test at significance level 5%) are highlighted in bold. This result corresponds to the top row in Figure 3.

| Data | Case | Val-only | Pretrain-val | Reweight | MW-Net | R-DIW | DIW | GIW |
|------|------|----------|--------------|----------|--------|-------|-----|-----|
| MNIST | (iii) | 78.94 (0.99) | 88.31 (1.02) | 82.03 (0.13) | 82.17 (0.32) | 81.73 (0.42) | 80.53 (1.23) | **95.16 (0.20)** |
| MNIST | (iv) | 79.86 (0.26) | 86.25 (0.95) | 77.78 (0.91) | 76.73 (0.63) | 77.11 (0.63) | 75.11 (1.43) | **90.94 (0.91)** |
| Color-MNIST | (iii) | 31.19 (0.65) | 80.40 (3.62) | 38.39 (1.03) | 39.26 (0.01) | 39.28 (0.03) | 39.50 (0.18) | **93.87 (0.19)** |
| CIFAR-20 | (iii) | 28.61 (0.66) | 42.79 (0.41) | 56.17 (0.29) | 57.90 (0.18) | 58.02 (0.35) | 55.78 (0.34) | **59.73 (0.47)** |

Table 4: Mean accuracy (standard deviation) in percentage on MNIST, Color-MNIST, and CIFAR-20 over the last ten epochs under support shift in case (iii) with DA baselines (5 trials). Best and comparable methods (paired *t*-test at significance level 5%) are highlighted in bold. This result corresponds to the bottom row in Figure 3.

| Data | Case | Val-only | Pretrain-val | CCSA | DANN | GIW |
|------|------|----------|--------------|------|------|-----|
| MNIST | (iii) | 78.94 (0.99) | 88.31 (1.02) | 94.37 (0.46) | 87.24 (1.58) | **95.16 (0.20)** |
| MNIST | (iv) | 79.86 (0.26) | 86.25 (0.95) | 88.76 (0.86) | 82.92 (1.89) | **90.94 (0.91)** |
| Color-MNIST | (iii) | 31.19 (0.65) | 80.40 (3.62) | 80.13 (5.90) | 64.34 (5.08) | **93.87 (0.19)** |
| CIFAR-20 | (iii) | 28.61 (0.66) | 42.79 (0.41) | **60.87 (0.39)** | 58.12 (0.44) | 59.73 (0.47) |

Table 5: Mean accuracy (standard deviation) in percentage on MNIST, Color-MNIST (C-MNIST), and CIFAR-20 over the last ten epochs under support-distribution shift in case (iii) with IW-like baselines (5 trials). Best and comparable methods (paired *t*-test at significance level 5%) are highlighted in bold. LN/CS is short for label noise/class-prior shift. This result corresponds to Figure 5.

| Data | Shift | Val-only | Pretrain-val | Reweight | MW-Net | R-DIW | DIW | GIW |
|------|-------|----------|--------------|----------|--------|-------|-----|-----|
| *MNIST* | LN 0.2 | 78.94 (0.99) | 86.34 (0.77) | 89.28 (0.64) | 82.29 (0.80) | 82.70 (0.76) | 80.12 (2.90) | **94.60 (0.22)** |
| | LN 0.4 | 78.94 (0.99) | 85.43 (0.97) | 84.73 (1.27) | 80.92 (0.52) | 81.19 (0.39) | 79.00 (0.33) | **93.74 (0.53)** |
| | CS 10 | 78.94 (0.99) | 86.43 (0.71) | 80.99 (0.67) | 79.76 (0.60) | 80.15 (0.38) | 79.73 (0.38) | **93.43 (0.40)** |
| | CS 100 | 78.94 (0.99) | 84.54 (1.61) | 80.65 (0.71) | 77.06 (0.71) | 76.53 (0.45) | 78.58 (0.96) | **91.52 (0.59)** |
| *C-MNIST* | LN 0.2 | 31.19 (0.65) | 60.21 (4.47) | 30.82 (1.04) | 40.00 (0.44) | 38.25 (0.46) | 38.04 (1.83) | **90.99 (0.61)** |
| | LN 0.4 | 31.19 (0.65) | 57.13 (5.48) | 27.26 (0.41) | 39.98 (0.06) | 39.08 (0.02) | 39.02 (0.50) | **90.09 (0.88)** |
| | CS 10 | 31.22 (0.69) | 87.63 (1.97) | 38.61 (0.62) | 39.16 (0.19) | 39.15 (0.19) | 39.18 (0.19) | **94.43 (1.87)** |
| | CS 100 | 31.22 (0.69) | 79.70 (3.29) | 37.70 (0.21) | 36.81 (0.10) | 36.75 (0.17) | 36.98 (0.23) | **88.34 (0.87)** |
| *CIFAR-20* | LN 0.2 | 29.18 (0.30) | 42.48 (0.55) | 47.01 (0.87) | 49.19 (0.31) | 50.13 (0.29) | 48.70 (0.36) | **54.19 (0.69)** |
| | LN 0.4 | 29.18 (0.30) | 39.46 (0.47) | 38.94 (1.10) | 43.29 (0.42) | 43.10 (0.29) | 43.23 (0.29) | **45.48 (0.36)** |
| | CS 10 | 28.45 (0.27) | 37.31 (0.28) | 42.25 (0.68) | 44.96 (0.21) | 45.19 (0.20) | 39.68 (0.14) | **49.17 (0.40)** |
| | CS 100 | 28.45 (0.27) | 36.55 (0.29) | 31.95 (0.77) | 34.66 (0.28) | 34.87 (0.12) | 26.39 (1.05) | **40.27 (2.17)** |

Table 6: Mean accuracy (standard deviation) in percentage on MNIST, Color-MNIST (C-MNIST), and CIFAR-20 over the last ten epochs under support-distribution shift in case (iii) with DA baselines (5 trials). Best and comparable methods (paired *t*-test at significance level 5%) are highlighted in bold. LN/CS is short for label noise/class-prior shift. This result corresponds to Figure 6.

| Data | Shift | Val-only | Pretrain-val | CCSA | DANN | GIW |
|------|-------|----------|--------------|------|------|-----|
| *MNIST* | LN 0.2 | 78.94 (0.99) | 86.34 (0.77) | 92.95 (0.20) | 87.33 (3.22) | **94.60 (0.22)** |
| | LN 0.4 | 78.94 (0.99) | 85.43 (0.97) | 89.83 (0.52) | 65.56 (8.31) | **93.74 (0.53)** |
| | CS 10 | 78.94 (0.99) | 86.43 (0.71) | 89.18 (0.33) | 73.23 (0.66) | **93.43 (0.40)** |
| | CS 100 | 78.94 (0.99) | 84.54 (1.61) | 72.67 (1.75) | 49.28 (0.00) | **91.52 (0.59)** |
| *C-MNIST* | LN 0.2 | 31.19 (0.65) | 60.21 (4.47) | 84.32 (4.57) | 60.65 (5.96) | **90.99 (0.61)** |
| | LN 0.4 | 31.19 (0.65) | 57.13 (5.48) | 80.32 (0.93) | 70.27 (9.09) | **90.09 (0.88)** |
| | CS 10 | 31.22 (0.69) | 87.63 (1.97) | 67.39 (3.07) | 49.65 (7.98) | **94.43 (1.87)** |
| | CS 100 | 31.22 (0.69) | 79.70 (3.29) | 64.15 (6.80) | 36.16 (5.80) | **88.34 (0.87)** |
| *CIFAR-20* | LN 0.2 | 29.18 (0.30) | 42.48 (0.55) | 52.49 (0.58) | 48.31 (0.44) | **54.19 (0.69)** |
| | LN 0.4 | 29.18 (0.30) | 39.46 (0.47) | 40.68 (0.70) | 42.29 (0.70) | **45.48 (0.36)** |
| | CS 10 | 28.45 (0.27) | 37.31 (0.28) | 46.17 (0.20) | 41.58 (0.34) | **49.17 (0.40)** |
| | CS 100 | 28.45 (0.27) | 36.55 (0.29) | 34.79 (0.35) | 33.90 (0.14) | **40.27 (2.17)** |

Table 7: Mean accuracy (standard deviation) in percentage on MNIST over the last ten epochs under support-distribution shift in case (iv) with IW-like baselines (5 trials). Best and comparable methods (paired $t$-test at significance level 5%) are highlighted in bold. LN/CS means label noise/class-prior shift. This result corresponds to the top row in Figure 11.

| Shift | Val-only | Pretrain-val | Reweight | MW-Net | R-DIW | DIW | GIW |
|---|---|---|---|---|---|---|---|
| LN 0.2 | 79.86 (0.26) | 85.08 (0.71) | 86.69 (0.80) | 78.16 (1.30) | 78.36 (1.08) | 75.96 (1.39) | **91.44 (0.85)** |
| LN 0.4 | 79.86 (0.26) | 83.26 (1.08) | 81.01 (1.73) | 75.60 (0.49) | 75.76 (0.56) | 72.12 (0.86) | **88.84 (0.97)** |
| CS 10 | 79.86 (0.26) | 83.44 (1.24) | 76.41 (1.28) | 74.67 (0.84) | 74.94 (0.59) | 74.75 (0.69) | **90.20 (0.66)** |
| CS 100 | 79.86 (0.26) | 82.09 (0.93) | 74.84 (1.10) | 71.42 (0.99) | 71.11 (0.54) | 73.66 (1.32) | **87.14 (0.51)** |

Table 8: Mean accuracy (standard deviation) in percentage on MNIST over the last ten epochs under support-distribution shift in case (iv) with DA baselines (5 trials). Best and comparable methods (paired $t$-test at significance level 5%) are highlighted in bold. LN/CS means label noise/class-prior shift. This result corresponds to the bottom row in Figure 11.

| Shift | Val-only | Pretrain-val | CCSA | DANN | GIW |
|---|---|---|---|---|---|
| LN 0.2 | 79.86 (0.26) | 85.08 (0.71) | 88.06 (0.48) | 85.32 (1.11) | **91.44 (0.85)** |
| LN 0.4 | 79.86 (0.26) | 83.26 (1.08) | 84.00 (0.70) | 58.02 (6.63) | **88.84 (0.97)** |
| CS 10 | 79.86 (0.26) | 83.44 (1.24) | 83.60 (0.53) | 64.30 (0.42) | **90.20 (0.66)** |
| CS 100 | 79.86 (0.26) | 82.09 (0.93) | 66.31 (0.92) | 50.06 (0.00) | **87.14 (0.51)** |

