# OpenReview forum: "Generalizing Importance Weighting to A Universal Solver for Distribution Shift Problems"
_NeurIPS.cc/2023/Conference — NeurIPS 2023 spotlight_

### Official Review · Reviewer_G7Ea · 2023-07-03

**Soundness:** 3 good
**Presentation:** 3 good
**Contribution:** 3 good
**Rating:** 8
**Confidence:** 5

**Summary:**

The paper studied importance weighting that is a fundamental technique in machine learning and data mining. It pointed out a serious issue of existing importance weighting methods that has not been realized or considered carefully --- the support of the training distribution may not fully enclose the support of the test distribution. Then, it proposed a generalized version of importance weighting that can solve the serious issue and has some nice theoretical properties.

**Strengths:**

The motivation is clear. People employ importance weighting where the importance weights are defined as pte(x,y)/ptr(x,y), and people assume pte(x,y)=0 whenever ptr(x,y)=0, to avoid unbounded importance weights. Consider the case that the support of the training distribution may not fully enclose the support of the test distribution. Since the importance weights are only used for the training distribution, even though they become ill-defined when pte(x,y)>0 and ptr(x,y)=0, importance weighting itself is still well-defined and it seems to be fine... However, this paper pointed out that in such a case the importance-weighting "identity" becomes an "inequality", which means that what we minimize for training will not be an approximation of the original risk and thus training may be biased. Indeed, this paper constructed a toy example (Figure 2 and Proposition 3) where training is still unbiased or totally biased. I like Proposition 3 very much that emphasizes what people have overlooked so far.

The idea is novel. It splits the validation data coming from the test distribution into two parts based on one-class classification and pretrained representation (note that pretraining is on the distribution-shifted training data and the proposed method is still end-to-end). Then, the original importance-weighting term is based on one part of the validation data and there is a second term based on the other part of the validation data. In such a way, what we minimize for training is an approximation of the original risk and thus training is still unbiased, as long as the split of validation data is not wrong. As far as I know, this idea or a similar idea has not been explored in importance-weighting research.

The experiments look very interesting. The Color-MNIST experiments (both the design and the results) are thought-provoking.

**Weaknesses:**

The experiments are all done on benchmark datasets at the MNIST/CIFAR scale and there is no big data experiment like ImageNet.

The biggest model in the experiments is ResNet-18 and there is no big model experiment involving WideResNet, DenseNet, or ViT.

**Questions:**

A point in the motivation should be clarified. As I mentioned above in "the motivation is clear", when the importance weights are ill-defined outside the training distribution, importance weighting itself is still well-defined. So have people not noticed the serious issue of importance weighting, or have they noticed it but thought it not a big deal and then ignored it? BTW, you only argued that "IW methods become ill-defined and problematic", which is not enough as an explanation. The importance weights may be ill-defined, but those ill-defined importance weights will not be used for training. As a result, IW methods are not ill-defined, but they are not good for training and may lead to poor trained classifiers. However, if we are lucky, they can lead to good trained classifiers, as shown in Figure 2. Am I right?

Before Proposition 3, you have assumed that "the classifier would transfer its knowledge about ptr(x,y) to pte(x,y) in the simplest manner". Why do you need this assumption, or can you explain the implicit meaning of this assumption? If I am not wrong, importance weighting can only train linear classifiers given such training and validation data.

In this paper, the definition of importance weighting is to use pte(x,y)/ptr(x,y) as the importance weights. There is a more general definition of importance weighting: weighting the loss function on training data to emphasize the loss on some or de-emphasize it on others with the objective of correcting the mismatch between the training and test distributions (see Corinna Cortes, Yishay Mansour, Mehryar Mohri, "Learning Bounds for Importance Weighting", NIPS 2010). This more general definition can include your "IW-like methods" as IW methods. You theoretically and empirically analyzed when and why IW methods can succeed/may fail. Can your results still hold for this more general definition?

I think importance weighting is not good at the case where the support of the training distribution may not fully enclose the support of the test distribution. This is the motivation of generalizing importance weighting in this paper. However, transfer learning and domain adaptation should also be good at the case which importance weighting is not good at and generalized importance weighting is good at. I found transfer learning or domain adaptation was not mentioned or discussed in this paper. So can you comment on their potential and compare their ability with generalized importance weighting?

**Limitations:**

Labeled validation data coming from the test distribution are required for estimating the importance weights and training. Such validation data may be hard to obtain in practice.

---

> ### Author Rebuttal · Authors · 2023-08-04
>
> Thanks for spending a lot of time reviewing our manuscript!
>
> ***1. Experiment scale***
>
> Please find our reply *"1. On the common concern about the experiment design"* in the top-level rebuttal above as well as the reply to Reviewer uQtu (https://openreview.net/forum?id=KmdlUP23qh&noteId=m5Q81fgcQF). In addition, do you have any suggestions for the type of support shift that we could simulate on ImageNet? We'd like to hear your advice.
>
> ***2. Have people not noticed the serious issue, or have they noticed it but thought it not a big deal and then ignored it?***
>
> We are not sure, but it might be the latter case. Before deep learning got so popular in the wild, even if there was support shift it must be *only a little* support shift, and it shouldn't be a big deal. Now, it's a different story. After being deployed in the wild, a model will see so many new concepts (i.e., classes or sub-classes) that it has *never seen during training*. Then, it can be a big deal.
>
> By the way, for the motivation, your understanding is perfectly correct.
>
> ***3. Why do you need this assumption, or can you explain the implicit meaning of this assumption?***
>
> Yes, as you said, "importance weighting can only train linear classifiers given such training and validation data". However, a richer function class is required to allow true risk minimization to train non-linear classifiers and recover the Bayes optimal classifier (i.e., there is *zero approximation error*). Otherwise, comparing the classifiers obtained by IW and true risk minimization, we cannot know that IW is as bad as random guessing.
>
> ***4. Can your results still hold for this more general definition?***
>
> It depends. The key point is *to what extent the validation data join the optimization of models*. When validation data directly join model training, an IW-like method should behave similarly to GIW (i.e., good performance); when validation data indirectly join model training,  it should behave similarly to the original IW (i.e., bad performance). For example, Reweight has a primary objective where the empirical risk on validation data is minimized to update the instance weights $w_i$ and a secondary objective (i.e., the constraint) where the empirical risk on weighted training data is minimized to update the model $\theta$ itself, so that only some but not all classifiers would be the candidates in the primary objective. In this way, the validation data don't directly join model training, and we can see from the experimental results in our manuscript that Reweight looks like DIW and R-DIW. We can also see that Meta-Weight-Net looks like Reweight, because it is roughly a parametric version of the non-parametric Reweight.
>
> ***5. Domain adaptation***
>
> Please find our reply *"2. About domain adaptation"* in the top-level rebuttal.
>
> Thanks again for your positive review and hope our rebuttal clarifies your major concerns.

---

> > ### Comment · Reviewer_G7Ea · 2023-08-15
> > **Thanks**
> >
> > Thank you for your feedback. My questions have been well answered, and I don't have any more questions. I think this paper is worth to be accepted.

---

> > > ### Author Response · Authors · 2023-08-15
> > >
> > > Dear Reviewer G7Ea,
> > >
> > > Thank you for the reply. It's great to hear that we have answered your questions.
> > >
> > > Authors

---

> > > ### Author Response · Authors · 2023-08-21
> > > **Sincere thanks**
> > >
> > > Thank you for taking your precious time and reviewing our submission. Unfortunately, there is still a non-responsive reviewer leaving a very negative review. You gave us a professional review with a high score, and we feel that you are an expert on our research topic. We would greatly appreciate your continued support of our submission during the reviewer-AC discussion.

---

### Official Review · Reviewer_jhs4 · 2023-07-04

**Soundness:** 4 excellent
**Presentation:** 3 good
**Contribution:** 3 good
**Rating:** 7
**Confidence:** 4

**Summary:**

This paper focuses on an important technique in the field of learning under distribution shift. Compared to previous importance weighting techniques, this paper consider more general scenarios: support shift. This paper first investigates how the previous techniques fail on general support-shift scenarios, then proposes a theoretical-justified objective to consider such scenarios. To this end, the whole pipeline is proposed based the objective. In the experiments, necessary empirical justification is included, verifying the effectiveness of the proposed method.

**Strengths:**

1.  The problem setting is interesting and important to the field. In the real world, cases (iii) and (iv) are very common. However, previous techniques do not consider such scenarios.

2. This paper contributes a principle to the field: what kind of objective is classifier-consistency when solving the general distribution-shift problem (i.e., the support-shift problem).

3. The experiments justify the proposed component. The selected baselines cover a lot of existing methods, which is very appreciated.


**Weaknesses:**

1. The four cases are clearly defined. However, potential practical scenarios can be justified better. In my opinion, test distribution’s support set containing training distribution’s support set might less happen in the current era (as researchers tend to train networks with a lot data). Case (iv) is the most common one, instead of case (iii).

2. In Figure 2, the word “unseen” will cause confusion. We indeed have validation data that come from the test distribution. Thus, it is better to use training data instead of seen/unseen data in Figure 2, or clearly show the decision boundary to avoid potential confusion.

3. It is not well-motivated why we need to introduce risk consistency/inconsistency from label-noise literature. It is a plain definition and should be motivated from its own.

4. Theorem 4 is important to justify the designed objective. However, the most important part, how estimation/observation of s influences the generalization ability is not explored. Necessary discussions should be considered to strengthen the theoretical contribution of this paper.

5. In the experiment, the considered dataset is relatively small. Tasks from partial domain adaptation problem can be used for validating the performance on case (iii). Some large training sets should be considered, e.g., ImageNet.

6. It is unclear how the performance of one-class classifier influences the performance of the proposed method. More experimental analysis should be considered towards this direction.



**Questions:**

1. It is not well-motivated why we need to introduce risk consistency/inconsistency from label-noise literature. It is a plain definition and should be motivated from its own.

2. Theorem 4 is important to justify the designed objective. However, the most important part, how estimation/observation of s influences the generalization ability is not explored. Necessary discussions should be considered to strengthen the theoretical contribution of this paper.

3. In the experiment, the considered dataset is relatively small. Tasks from partial domain adaptation problem can be used for validating the performance on case (iii). Some large training sets should be considered, e.g., ImageNet.

4. It is unclear how the performance of one-class classifier influences the performance of the proposed method. More experimental analysis should be considered towards this direction.



**Limitations:**

This paper does not have potential negative societal impact to be addressed. Paper’s technical limitations can be found in the above comments.

---

> ### Author Rebuttal · Authors · 2023-08-04
>
> Thanks for your constructive comments!
>
> For *Questions 2 to 4*, please find our replies in the top-level rebuttal above as well as the reply to Reviewer uQtu (https://openreview.net/forum?id=KmdlUP23qh&noteId=m5Q81fgcQF). In addition, do you have any suggestions for the type of support shift that we could simulate on ImageNet? We'd like to hear your advice.
>
> Next, for *Question 1* and partially *Question 2*, our Theorem 4 studies risk consistency, where *neither estimation nor optimization is involved*. This made theoretical analysis easier and was exactly why we introduced risk consistency.
>
> The theoretical analysis of the quality of estimating $s$ is more difficult than the theoretical analysis of the quality of estimating $w_i$, while even the latter was not included in previous papers such as Reweight or DIW (note that the convergence analysis in Reweight was for its optimization convergence). This is simply because *representation learning* plays an extremely important role (which was the sole motivation of DIW) but its own quality is actually not guaranteed *under joint shift* in the worst case.
>
> On the other hand, if the model is a linear model (including linear-in-input and linear-in-parameter models), the quality of estimating $s$ and that of estimating $w_i$ can be guaranteed by the existing theory of linear *one-class classification* [1,2] or *positive-unlabeled classification* [3,4] and that of linear *distribution matching* [5] or *density ratio estimation* [6]. Since we are only interested in *IW for deep learning*, we left the aforementioned theoretical analyses as open problems.
>
> Nevertheless, we have provided some empirical analyses in the supplementary material (see Figure 8 in https://openreview.net/attachment?id=KmdlUP23qh&name=supplementary_material) and the rebuttal, which illustrated the concerned quality in the average case.
>
> Thanks again for your positive review and hope our rebuttal clarifies your major concerns.
>
> [1] Kernel Methods for Pattern Analysis (Chapter 7), Cambridge University Press 2004.
>
> [2] Advances in Statistical Models for Data Analysis (Chapter 24), Springer 2015.
>
> [3] Semi-supervised novelty detection, JMLR 2010.
>
> [4] Machine Learning from Weak Supervision: An Empirical Risk Minimization Approach (Chapter 4), The MIT Press 2022.
>
> [5] Dataset Shift in Machine Learning (Chapter 8), The MIT Press 2008.
>
> [6] Density Ratio Estimation in Machine Learning (Chapters 13 to 16), Cambridge University Press 2012.

---

> > ### Comment · Reviewer_jhs4 · 2023-08-11
> >
> > Most of my concerns have been resolved.

---

> > > ### Author Response · Authors · 2023-08-15
> > >
> > > Dear Reviewer jhs4,
> > >
> > > Thank you for the reply. It's great to hear that we have answered your questions.
> > >
> > > Authors

---

> > > ### Author Response · Authors · 2023-08-21
> > > **Sincere thanks**
> > >
> > > Thank you for taking your precious time and reviewing our submission. Unfortunately, there is still a non-responsive reviewer leaving a very negative review. You gave us a professional review with a high score, and we feel that you are an expert on our research topic. We would greatly appreciate your continued support of our submission during the reviewer-AC discussion.

---

### Official Review · Reviewer_uQtu · 2023-07-06

**Soundness:** 2 fair
**Presentation:** 2 fair
**Contribution:** 2 fair
**Rating:** 6
**Confidence:** 2

**Summary:**

The paper introduces a Generalized Importance Weighting (GIW) method designed to address distribution shift problems, where the support of the probability density function varies between training and test data. The authors demonstrate that GIW outperforms existing methods in scenarios where the test support is wider or partially overlaps with the training support.

**Strengths:**

**Originality**: The paper introduces a novel method called Generalized Importance Weighting (GIW) that extends the traditional Importance Weighting (IW) method.

**Quality**: The paper is well-structured and provides a thorough analysis of why IW may fail in certain cases and how GIW addresses these issues.

**Clarity**: This paper is well-written and easy to follow.


**Weaknesses:**

- This paper evaluates GIW on a few small datasets. However, it would be beneficial to evaluate the method on a large and complex dataset such as ImageNet.   This would provide a more comprehensive understanding of the performance and applicability of GIW.

-  All experiments in this paper are based on small models LetNet-5 and resnet-18. It would be beneficial to evaluate the method on other modern architectures such as vision transformers and VGG.  This would provide a more comprehensive understanding of whether the proposed method is sensitive to model architectures.


**Questions:**

- Can the proposed GIM perform well on other model architectures such as VGG/Vision Transfomer and large real-world datasets such as ImageNet?

- The authors utilize O-SVM to split D_v to IT and OOT validation data. How to make sure the O-SVM correctly split the D_v set and how does the quality of O-SVM splitting affect the performance of the  proposed GIW?

**Limitations:**

The authors adequately addressed the limitations.

---

> ### Author Rebuttal · Authors · 2023-08-03
>
> Thanks for your constructive comments!
>
> The experiment scale is concerned by 3 reviewers (i.e., 3 reviewers commented on the size of datasets and 2 commented on the size of models), so please find our reply *"1. On the common concern about the experiment design"* in the top-level rebuttal above. In addition, do you have any suggestions for the type of support shift that we could simulate on ImageNet? We'd like to hear your advice.
>
> For O-SVM related issues, please find our reply *"3. About one-class SVM"* in the top-level rebuttal as well.
>
> As far as we know, there are only very few instance weighting/reweighting methods that can properly deal with joint shift and are compatible with deep learning at the same time: Learning to Reweight (ICML 2018), Meta-Weight-Net (NeurIPS 2019), and Dynamic Importance Weighting (NeurIPS 2020). All of them conducted experiments at the MNIST/CIFAR-10 scale. This is because researchers don't use those high-quality benchmark datasets as they are, but many experiment setups are generated to simulate distribution shift problems. For each experiment setup, we used 5 random seeds to simulate distribution shift (a key point is that *we need to simulate support shift at the same time*), and hence it can easily take a lot of time for training deep networks even though they are not big deep networks. Hyperparameter tuning is also needed, which makes the experimenting time even longer than what people usually expect. We understand it shouldn't be an excuse that in this small research area previous researchers did so and then we are doing the same. However, the current experiment scale may already be a limit beyond which we cannot practically afford.
>
> Fortunately, *GIW is NOT sensitive to the model architecture* if we replace the models with smaller deep networks. The accuracy of all methods would be lower but the overall trend would not change --- GIW would still generally outperform the baselines. This has been verified in our preliminary experiments.
>
> Thanks again for your positive review and hope our rebuttal clarifies your major concerns.

---

> > ### Comment · Reviewer_uQtu · 2023-08-16
> >
> > Thank you for your rebuttal. Most of my concerns have been resolved. Therefore, I increase my score to 6.

---

> > > ### Author Response · Authors · 2023-08-16
> > >
> > > Dear Reviewer uQtu,
> > >
> > > Thank you very much. We are happy to hear that most of your concerns have been resolved.
> > >
> > > Authors

---

> > > ### Author Response · Authors · 2023-08-21
> > > **Sincere thanks**
> > >
> > > Thank you for taking your precious time and reviewing our submission. Unfortunately, there is still a non-responsive reviewer leaving a very negative review. You gave us a professional review with a high score, and we feel that you are an expert on our research topic. We would greatly appreciate your continued support of our submission during the reviewer-AC discussion.

---

### Official Review · Reviewer_UAYX · 2023-07-09

**Soundness:** 3 good
**Presentation:** 3 good
**Contribution:** 3 good
**Rating:** 6
**Confidence:** 3

**Summary:**

The paper considers a distribution shift problem where the test distribution $D_\text{te}$ is different from the training distribution $D_\text{tr}$.
Specifically, the learner is given training data sampled according to $D_\text{tr}$ and (typically a much smaller size of) validation data sampled according to $D_\text{te}$, and is required to output a classifier with small generalization error (measured by $D_\text{te}$).
The paper first exemplifies a particular pair of $D_\text{tr}$ and $D_\text{te}$ such that the existing method called the importance weighting fails, and then
improves the method to better handle such hard cases.

**Strengths:**

Give a hard example where the existing method has significantly poor performance but the proposed method has reasonable performance.

Revise the objective function so as to be risk-consistent, and then propose its empirical version to be minimized, which gives a learning algorithm that outperforms the existing method in experiments.

**Weaknesses:**

First of all, Theorem 4, claimed as the main theorem, is just a simple rewrite of the generalization risk and thus straightforward.
Moreover, the theorem does not give any justification of the proposed method. Instead, the consistency of the estimator (the empirical objective) should be investigated. (Generally, the notion of consistency is defined for esitimators.)
I do not think that the expectation of $\hat J_G(f)$ coincides $R(f)$.
---
My concern is addressed thru the discussion with the authors.

**Questions:**

I wonder why the authors divide the problems into four cases according to the supports of $D_\text{tr}$ and $D_\text{te}$.
Apparently, any distribution $D$ with support $S$ is empirically indistinguishable from a distribution $D'$ with support $S'$ if $D$ (and $D'$, resp.) has very small but non-zero probability mass for all $x \notin S'$ ($x \notin S$, resp.).

**Limitations:**

It seems to be impossible to establish a perfectly risk-consistent estimator in general. So we may need to put some assumptions about the difference between $D_\text{tr}$ and $D_\text{te}$ and investigate to what extent an estimator is risk-consistent.

---

> ### Author Rebuttal · Authors · 2023-08-06
>
> Thank you for reviewing but there seem to be some serious factual misunderstandings. We will clarify the issues one by one.
>
> ***1. "generally, the notion of consistency is defined for estimators"***
>
> Here, we guess estimators meant the empirical objectives from the previous sentence. This claim is NOT true in machine learning, especially in weakly supervised learning.
>
> Let's go back to the origin of consistency. Ronald Fisher created the notion of consistency in 1922 for *point estimation* in *parametric statistics*. This consistency asserts that "if the estimator were calculated using the entire population rather than a sample, the true value of the estimated parameter would be obtained". Let $\theta$ be the fixed but unknown parameter for $F_\theta$ and $T$ be an estimator that $\hat{\theta}=T(\hat{F}\_n)$. Then, $T$ is said to be consistent if $T(F\_\theta)=\theta$ or $T(\lim\_{n\to\infty}\hat{F}\_n)=\theta$. The definition is NOT $\lim\_{n\to\infty}T(\hat{F}\_n)=\theta$, even though they are equivalent in parametric statistics.
>
> They are not equivalent and Fisher's is stronger than the commonly used one in *non-parametric statistics*. This is particularly true in ML where the target of convergence (an optimal classifier) is defined by optimization rather than being a fixed but unknown function: even in discriminative learning, a deterministic function cannot generate the observed labels. People care more about learned classifiers (risk minimizers) than risk estimators. *A bad risk estimator may still lead to a good classifier*, for example, $\hat{R}(f)=2R(f)$ and $\hat{R}(f)=R(f)^2$. Therefore, ML people talk about *consistency of classifiers rather than consistency of risk estimators*.
>
> ***2. Types of guarantee in ML and WSL***
>
> "Fisher consistency" is a bit ambiguous in ML: the original definition became *classification-calibration* [1] and the limit definition became *strong consistency*. In the former case, the true risk minimizer is the true error minimizer (error is risk with 0-1 loss), where the *surrogate loss* is consistent. In the latter case, the empirical risk minimizer approaches the Bayes optimal classifier (using a surrogate loss), where the empirical risk minimizer is consistent. There is also *weak consistency* replacing the Bayes optimal classifier with the optimal classifier within a *function class*. Sorry that we cannot give detailed difference between them due to limited space. In ML, the term consistency usually means weak consistency.
>
> In WSL, training data are not from the test distribution, and thus we need more types of guarantee (see our paper, [2], and the references therein). First, *risk consistency* (RC) means that we can express the true risk as an expectation over the training distribution and this holds *for every classifier*, which is the most fundamental guarantee in WSL. Second, when a RC obj can be naively approximated, we enjoy *unbiased risk estimators* (URE) that can easily lead to statistical consistency (SC). However, the new obj may be complicated (e.g., a max operator in *non-negative correction* [3] or a weight estimator in IW); sometimes, a risk estimator is asymptotically unbiased and we can still have SC. Third, *classifier consistency* (CC) can be regarded as the WSL version of classification-calibration/the original Fisher consistency.
>
> As classification-calibration is the weakest condition in supervised learning [1] (which should always be satisfied), CC is also the weakest condition in WSL. RC and SC are both stronger than CC and *they cannot imply each other*, for example, $J(f)=R(f)^2$ gives SC empirical risk minimizers but it's not RC. URE is stronger than RC and SC. Sorry that we cannot give detailed difference between them due to limited space but we can discuss it later.
>
> [1] Convexity, classification, and risk bounds. JASA 2006.
>
> [2] Machine Learning from Weak Supervision: An Empirical Risk Minimization Approach. The MIT Press 2022.
>
> [3] Positive-unlabeled learning with non-negative risk estimator. NeurIPS 2017.
>
> ***3. Our theoretical contributions***
>
> We proved that the standard IW is not RC and may be even not CC in some cases, and then proved that GIW is always RC in all cases.
>
> ***4. "I do not think that the expectation of $\hat{J}_G(f)$ coincides $R(f)$"***
>
> We never claimed so. Instead, we explicitly mentioned that $\hat{J}_G(f)$ is not URE (see Footnote 3 on Page 7). As far as we know, *there doesn't exist any URE in IW*. On the other hand, $\hat{J}_G(f)$ can be consistent (and asymptotically unbiased) if its weight estimator has no approximation error. That's why we rely on non-parametric weight estimation.
>
> ***5. "the consistency of the estimator (the empirical objective) should be investigated"***
>
> See Footnote 3 on Page 7. See also the reply to Reviewer jhs4 (https://openreview.net/forum?id=KmdlUP23qh&noteId=6CQeO3yhmm) for a related discussion.
>
> ***6. "I wonder why the authors divide the problems into four cases"***
>
> Please don't mix up the theoretical and empirical perspectives. The theoretical contributions hold even if support shift is only a little. On the other hand, the experiment design considered the cases that the training support is 40% (for sub-class shift) or 33% (for color shift) of the test support, where *support shift matters* for IW. Two-sample tests aren't good at detecting support shift even if it's huge. That's exactly why we proposed GIW.
>
> ***7. "we may need to put some assumptions about the difference between $D_{tr}$ and $D_{te}$"***
>
> Learning under distribution shift requires either *distributional assumption* or test-distributed validation data. For joint shift, some test-distributed validation data are given and thus no additional distributional assumption is needed. If a distributional assumption is given such as $p(x)$ doesn't change, joint shift will reduce to the 4 easier types. The only assumption here is that the training and test distributions *must have some overlap*.

---

> > ### Author Response · Authors · 2023-08-15
> > **Slightly more discussion**
> >
> > We'd like to talk slightly more about why we divided the set of distribution shift problems into four cases according to support shift.
> >
> > As you pointed out, we cannot empirically detect any small support shift; in fact, even if the support shift is huge, we cannot empirically detect it either (under joint shift). The only empirically distinguishable support shift is covariate shift ("a learning based hypothesis test for harmful covariate shift", ICLR 2023), where the hypothesis test method is learning-based and specially designed, so that it doesn't work for more complex joint shift. Therefore, dividing distribution shift into four cases according to support shift is not a bug, it's a feature of our submission.
> >
> > The proposed GIW is a strictly more general case of the standard IW and it's safer to be used when we're not sure whether the problem is Case 1/2 or it's Case 3/4 --- even if we can empirically distinguish the four cases, we don't have/need to do so.

---

> > > ### Author Response · Authors · 2023-08-20
> > >
> > > We hope everything is going well in your work and life. We can understand you are extremely busy, but can you take a look at our rebuttal? It would take only 5 to 10 minutes. Thanks!

---

> > > > ### Comment · Area_Chair_9Cnm · 2023-08-20
> > > > **Reviewer UAYX's comments**
> > > >
> > > > It is unfortunate that, for some unknown reasons, Reviewer UAYX was not able to respond to the authors during the reviewer-authors discussion phase. However, after reading the authors' response, I think the authors have done a good job clarifying the concerns regarding the consistency of the estimator and the proposed four case studies.
> > > >
> > > > If Reviewer UAYX still has any additional concerns during the remaining time, please utilize this time to engage further with the authors. Thanks.
> > > >
> > > > AC

---

> > > > > ### Author Response · Authors · 2023-08-20
> > > > >
> > > > > Dear AC,
> > > > >
> > > > > We sincerely thank you very much for your reaction! It is truly unfortunate that for some unknown reasons Reviewer UAYX cannot respond to our rebuttal to his or her review. Anyway, we hope that everything goes well in his or her work and life.
> > > > >
> > > > > Authors

---

> > > > > > ### Comment · Reviewer_UAYX · 2023-08-21
> > > > > >
> > > > > > I'm very sorry for the late resonse. I've been extremely busy these days.
> > > > > > I went through the rebuttal by the authors and now I understand that the authors consider Fisher's consistency.
> > > > > > (Honestly I'm unfamiliar with the notion.)  So, I can raise the rating to "weak accept".
> > > > > > Stil, I believe that it's better to investigate the consistency of $\hat{J}_G(f)$ from the algorithmic point of view.
> > > > > > You claimed in footnote 3 in page 7 (sorry, I missed it) that $\hat {J}_G(f)$ is statistically consistent. Then I wonder why you do not make this claim as the (main) theorem.
> > > > > > Anyway I think we have no time for further discussions. Sorry again for the late comment.

---

> > > > > > > ### Author Response · Authors · 2023-08-21
> > > > > > >
> > > > > > > Dear Reviewer UAYX,
> > > > > > >
> > > > > > > It's really great to hear your feedback! We will explain why we didn't make *statistical consistency* (we will use consistency for short in this reply) a second main theorem about GIW.
> > > > > > >
> > > > > > > The consistency of training with IW relies on two parts: the consistency of *weight estimation* and the consistency of *weighted classification*. Consider the second part by assuming we already have the first part. As long as the importance weight is upper bounded by a finite constant, the consistency of weighted classification will be almost the same as that of the standard ERM, where the *estimation error bound* of training with IW would be the est err bound of ERM times the above upper bound of importance weight times 2 (here, the bound is further doubled since the empirical risk estimator is only consistent but not unbiased, and *symmetrization* is not as tight as before). Thus, the convergence rate would be the same.
> > > > > > >
> > > > > > > The difficulty is the first part. Here, it is truly an estimation problem rather than an optimization problem, and the target is not defined by optimization like in the second part. The target must be the ratio of test density over training density, namely $p_{te}(x,y)/p_{tr}(x,y)$. As a consequence, when talking about the consistency of weight estimation or estimated weights, we have to consider *not only estimation error but also approximation error*. In the literature, we have upper bounds of the difference between estimated importance weights and true importance weights (for example, MSE between them). However, the upper bounds don't really converge to zero when the number of data approach infinity, due to the existence of approximation error. When deep models are used and *representation learning* is involved, there is another key issue: the *feature extractor* $\pi:(x,y)\mapsto z$ is obtained from representation learning, and $p_{te}(x,y)/p_{tr}(x,y)=p_{te}(z)/p_{tr}(z)$ only if $\pi$ is *invertible* (there are two other trivial assumptions, and "only if" becomes "if and only if" with all three assumptions). This assumption is really non-trivial but it is only needed for theoretical purposes. We tried to avoid introducing this assumption, because the practical algorithms don't require to use invertible deep networks or so-called *deep invertible networks*.
> > > > > > >
> > > > > > > Fortunately, IW methods have an amazing property: *weighted classification works as long as we have roughly good estimated importance weights*. For example, we know the analytic solution of weights under class-prior shift, and if we assume that the true label is deterministic (which is approximately true for MNIST and CIFAR-10) we also know the analytic solution of weights under class-posterior shift. Intuitively, under class-prior shift, all instances belonging to majority classes should have a very small weight and all instances belonging to minority classes should have a very large weight; under class-posterior shift, all mislabeled instances should have a zero weight and all correctly labeled instances should have a clearly non-zero weight. We can train fairly good classifiers as long as the estimated weights can tell the difference between majority and minority classes or between mislabeled and correctly labeled instances. In fact, in optimization for machine learning, a true (global or local) minimizer with zero gradient may not necessarily be better than a reasonably good solution with near-zero gradient near the true minimizer. The situation of weight estimation is the same, where the true importance weights may not necessarily be better than some reasonably good estimated weights near the true weights, since we have finite data for weighted classification and *what we care most is weighted classification but not weight estimation*. That was also why researchers here don't care too much the consistency of weight estimation.
> > > > > > >
> > > > > > > We didn't have enough space to include the above discussion in our original submission. We omitted it because we thought our readers should mainly be deep learning researchers who may not care about this issue. However, we will try our best to include the discussion into our final submission, to avoid any potential confusion.
> > > > > > >
> > > > > > > Hope our explanation can clarify your concern and also hope you can take care of yourself during your extremely busy days! When you have some time, as you have promised, hope you can remember to "raise the rating to weak accept".
> > > > > > >
> > > > > > > Authors

---

### Author Rebuttal · Authors · 2023-08-03

***1. On the common concern about the experiment design***

We sincerely thank all the reviewers. First of all, since 3 reviewers asked for additional experimental comparisons of the proposed method against the baseline methods on ImageNet, we'd like to explain why it is difficult to do so. We will try our best to provide such results later.

**(a) Computational efficiency** All existing IW and IW-like methods are slow. It would be difficult to compare GIW with the baselines on large-scale datasets (we are using V100). The inner optimization of GIW (indeed DIW [1]), namely *kernel mean matching*, needs to be solved by a standard QP solver. However, convex programming solvers (including CPLEX, Gurobi, CVX, and CVXOPT) do not support GPUs and have no plan to do so. This is because they focus on sparse linear algebra that should be MIMD rather than SIMD. As a result, the inner optimization (obtaining importance weights) is done on CPUs and the outer optimization (training deep networks) is done on GPUs, where moving data between CPUs and GPUs is really time-consuming. Moreover, IW-like methods based on bilevel optimizations are also very slow due to complicated technical reasons. We monitored the training time of CIFAR-20 experiments with ResNet-18. On V100, GIW took 27.89s per epoch and Reweight [2] took 67.51s per epoch, respectively. This baseline is even slower than GIW and thus it is difficult to train big models on ImageNet for both of them and compare their accuracy.

Note that even if we replace KMM with similar methods like *least-squares importance fitting* [3], we can still not get rid of a QP solver. We may hand-crafted a rough QP solver that can run on GPUs, but it is already beyond the scope of the current paper. What we are extending is the objective function of IW but not the implementation of DIW because the plug-and-play weight estimator in GIW is not limited to DIW.

[1] Rethinking importance weighting for deep learning under distribution shift. NeurIPS 2020.

[2] Learning to reweight examples for robust deep learning. ICML 2018.

[3] A Least-squares Approach to Direct Importance Estimation. JMLR 2009.

**(b) Problem difficulty** As mentioned in our introduction (see also the ICLR keynote by Masashi Sugiyama), joint shift is much harder than the other 4 types of distribution shift. Any algorithm must be very careful when handling joint shift. Though the experiments simulated 2 easier types, GIW and the baselines didn't know it, and we cannot ask an algorithm to identify the type of easier shift since the joint shift may not fall into the 4 easier types in practice. Given that the problem of interest is so hard, we shouldn't expect too much for GIW or the baselines. Note that CIFAR-20 and CIFAR-100 share the same instances with two-level labels, making it a natural test bed for joint shift in case 3/4. We'll study how to simulate the same case on ImageNet, but it's indeed non-trivial.

***2. About domain adaptation***

**(a) Conceptual difference** DA can refer to either certain problem settings or the corresponding learning methods (or both). There are *Supervised DA* and *Unsupervised DA* in DA research. SDA was proposed in 2006 and UDA was proposed in 2007; UDA is now much more impactful than SDA (we don't see new powerful SDA methods in recent years). Regarding DA as problem settings, SDA is exactly the same as joint shift and UDA is fairly similar to covariate shift, where distribution shift research has a slightly longer history (dated back to 2000). Regarding DA as learning methods, the philosophy of both SDA and UDA is to find good representations to link the source and target domains and transfer knowledge from the source domain to the target domain, totally different from the philosophy of IW. In fact, *importance sampling* as a sister of IW can be dated back to 1978 in statistics and 1949 in statistical physics.

Please note that the main focus of this paper is conceptually extending IW and making it more applicable to harder problems, rather than finding better solutions for the harder problems. That's why we didn't focus on comparing GIW with DA.

**(b) Empirical results** We added 2 representative DA baselines: *classification and contrastive semantic alignment* (CCSA) [4] (for SDA, 700+ citations) and *domain-adversarial neural network* (DANN) [5] (for UDA, 7000+ citations). The experimental results are given as Tables 1&2 in the attachment, where GIW generally outperformed CCSA and DANN. Specifically, when there was no distribution shift other than support shift, CCSA was as good as (or slightly better than) GIW, since this is what SDA is good at and there is nothing to do with DIW inside GIW. When there was also class-posterior shift (label noise) or class-prior shift, CCSA became confused, since this is what IW is good at and the source domain can mislead the learned representation of CCSA. DANN was even worse than CCSA, simply because it didn't use target domain labels. To sum up, IW and DA have different philosophies and there are situations that GIW is good at but CCSA and DANN cannot properly handle.

[4] Unified deep supervised domain adaptation and generalization. ICCV 2017.

[5] Domain-adversarial training of neural networks. JMLR 2016.

***3. About one-class SVM***

As mentioned in our paper, while there are more advanced one-class classification, the O-SVM is already good enough for the purpose (see Figure 8 in Appendix C.1). This is because *the underlying representation learning is very powerful*. To further investigate the negative impact of split errors, we added an ablation study by randomly flipping the IT/OOT data into the OOT/IT part. From Table 3 in the attachment, we can observe that GIW is quite robust to split errors and that OOT-->IT is more problematic than the other direction. This is because OOT-->IT makes the empirical risk estimator of GIW more similar to that of the standard IW.

---

### Comment · Area_Chair_9Cnm · 2023-08-11
**Reviewer-author discussion**

Dear Reviewers,

Please take a moment to read the authors' responses. Your insights and feedback are crucial in ensuring a comprehensive evaluation. Thanks.

AC

---

### Decision · Program_Chairs · 2023-09-21

**Decision:**

Accept (spotlight)

**Comment:**

The paper has a clear explanation of importance weighting's assumptions and its identification of biases in scenarios where training and test distribution supports differ. The reviewers highlight the novelty of the proposed idea that splits validation data using one-class classification and pretrained representation to maintain unbiased training. The Generalized Importance Weighting (GIW) method's effectiveness in addressing distribution shift problems is emphasized.